# Acute regulation of murine adipose tissue lipolysis and insulin resistance by the TGFβ superfamily protein GDF3

Nagasuryaprasad Kotikalapudi [1,5], Deepti Ramachandran [1,5], Daniel Vieira [1], William B. Rubio[1], Gregory R. Gipson [2], Luca Troncone [3], Kylie Vestal [2], David E. Maridas[4], Vicki Rosen [4], Paul B. Yu [3], Thomas B. Thompson[2] & Alexander S. Banks [1] ✉

TGFβ superfamily proteins can affect cellular differentiation, thermogenesis, and fibrosis in mammalian adipose tissue. Here we describe a role for Growth Differentiation Factor 3 (GDF3) on mature adipocyte biology. We find inducible GDF3 loss of function in obese adult mice leads to reduced lipolysis, improved glucose tolerance, and reduced glycemic variability. The effects on lipolysis are driven by lower levels of β3-adrenergic receptor, decreased cAMP and PKA signaling. GDF3 is an ALK5, ALK7, ACVR2A and ACVR2B agonist and also a BMPR2 antagonist. Unlike ALK7 or activin E knockouts, acute GDF3 loss of function does not affect body weight or energy balance but significantly improves metabolic health. These results suggest that blocking GDF3 can improve metabolic health independent of body weight and food intake, an intriguing new model for developing anti-diabetic therapies. Together these results provide much-needed clarity to both the molecular pathways involved in GDF3 signaling and its physiological effects.

The transforming growth factor β (TGFβ) superfamily comprises TGFβ proteins, activins, growth differentiation factors (GDFs), and bone morphogenetic proteins (BMPs). These proteins are well-characterized in the context of metabolism and have diverse functions in regulating appetite, lipid metabolism, and glucose homeostasis. TGFβ superfamily proteins signal through combinations of heteromeric type I and II receptor complexes and these combinations vary by cell type and/or environmental signals. Upon ligand binding to receptor, the primary signal transducers are the SMAD proteins[1,2]. These proteins are important in maintaining overall metabolic homeostasis.

Growth Differentiation Factor 3 (GDF3) is a member of the TGFβ superfamily. GDF3 is highly expressed in stem cell populations and helps maintain pluripotency. However, Gdf3 mRNA levels decline sharply after differentiation[3,4]. Due to the critical role of GDF3 in embryonic development, mice with germline deletion of Gdf3 exhibit decreased survival and developmental abnormalities. While these models are protected from diet-induced obesity, the mechanistic basis for these phenotypes is challenging to interpret[3,5–7].

Considerable controversy exists surrounding what role GDF3 plays in differentiated cells and further clarification is needed[8,9]. Like other members of the TGFβ superfamily, GDF3's function may be highly cell-type dependent. In adult humans and mice, Gdf3 mRNA levels significantly increase in obesity, aging, inflammation, and following an ischemic event[10–14]. To understand the role of GDF3 in regulating these diseases, we need to understand its biological functions and mechanisms.

The nuclear hormone receptor PPARγ controls GDF3 expression[12]. Tissue-specific PPARγ loss of function leads to increased Gdf3 mRNA

[1]Division of Endocrinology, Diabetes and Metabolism, Beth Israel Deaconess Medical Center and Harvard Medical School, Boston, MA, USA. [2]Department of Molecular & Cellular Biosciences, University of Cincinnati College of Medicine, Cincinnati, OH, USA. [3]Cardiovascular Research Center, Massachusetts General Hospital and Harvard Medical School, Boston, MA, USA. [4]Department of Developmental Biology, Harvard School of Dental Medicine, Boston, MA, USA. [5]These authors contributed equally: Nagasuryaprasad Kotikalapudi, Deepti Ramachandran. ✉e-mail: asbanks@bidmc.harvard.edu

expression and conversely, activation of PPARγ with thiazolidinediones suppresses Gdf3 expression[11,12]. Of note, GDF3 levels are also decreased by mutating PPARγ so that it cannot be phosphorylated at the 273[rd] amino acid (Serine S273, a post-translational modification that positively correlates with increasing weight gain and obesity)[11,15–17]. In each instance, high GDF3 levels positively correlate with insulin resistance, while low GDF3 levels are associated with insulin sensitivity. Moreover, exogenously increasing GDF3 expression with adeno-associated virus (AAV) is sufficient to make even lean mice moderately insulin resistant[11]. These findings suggest that the elevated GDF3 levels in obesity may contribute to insulin resistance, positioning GDF3 as a candidate target gene to develop therapies for type 2 diabetes, obesity, or other indications.

Conflicting research exists about GDF3's signaling mechanism. Unlike nearly all TGFβ superfamily proteins, mature GDF3 lacks a highly conserved cysteine residue, likely impacting covalent homodimerization[18]. Some studies suggest the missing GDF3 cysteine promotes inhibitory GDF3/BMP heterodimerization with BMP signaling ligands and antagonizes BMP receptor signaling[19]. However, Gdf3 is most commonly associated with its role as a weak endogenous ligand for the ALK7 receptor[3,5,6,11,13,20,21]. ALK7 loss of function variants in humans are associated with decreased risk for developing type 2 diabetes and decreased omental fat mass[22]. This finding has been examined in mice, where global ALK7 deletion, adipose-tissue specific ALK7 knockout (ALK7 KO), or an ALK7 neutralizing antibody all lead to decreased body weight and fat mass. However, these animals also demonstrate a paradoxical effect of increased adipose tissue lipolysis and insulin resistance despite lower body weights[20,23,24]. These effects of lower fat mass, yet insulin resistance are phenocopied by mice with loss or inhibition of another ALK7 ligand, activin E[25].

This study aims to investigate the impact of GDF3 and associated metabolic effects on diet-induced obesity in mice. We conducted the study using inducible Gdf3 knockout mice to examine the effects of GDF3 loss of function after the onset of diet-induced obesity. Our findings show that mice with lower levels of GDF3 show improved glucose tolerance due to reduced levels of lipolysis. These data complement earlier studies demonstrating insulin resistance following Gdf3 overexpression. Loss of GDF3 as an ALK7 receptor agonist should produce lower fat mass and insulin resistance. Instead, we find that GDF3 loss of function has the opposite effect–lack of GDF3 in obese mice decreases lipolysis and promotes insulin sensitivity without affecting body weight. Conversely, GDF3 gain of function increases rates of lipolysis and insulin resistance. We find that while GDF3 may signal through the ALK7 (ACVR1c) type 1 receptor, it also strongly activates another type 1 receptor ALK5 (TGFBR1). We also identified GDF3 binding to three type II receptors, BMPR2, ACVR2A, and ACVR2B. Detailed investigation of GDF3 signaling revealed that GDF3 can simultaneously inhibit BMP-derived SMAD1/5/8 signaling and stimulate activin-like SMAD2/3 signaling in vitro, ex-vivo, and in vivo. Overall, our study provides new insights into the role of GDF3 in maintaining metabolic homeostasis.

## Results

### Acute GDF3 loss of function improves metabolic health in mice with diet-induced obesity and leads to reduced lipolysis

In adult mice, GDF3 is highly expressed in obese adipose tissue[11,13,26,27]. To investigate the impact of acute GDF3 loss of function in adult mice, we generated mice encoding a floxed Gdf3 allele (Gdf3[fl/fl]) and crossed these to mice encoding Rosa-Cre[ERT2][28]. This approach allowed us to conditionally delete Gdf3 with tamoxifen-induced Cre recombinase after the development of obesity (Fig. 1A and Supplementary Fig. 1A). Gdf3[fl/fl]::Rosa-Cre[ERT2/–] and Gdf3[fl/fl] mice were fed high-fat diet (HFD) for 8 weeks, and then all mice were administered tamoxifen to generate whole-body Gdf3 knockout mice (Gdf3[KO]) with acute deletion of Gdf3 and their littermate controls (Gdf3[fl/fl]). Following 4 weeks to washout

any residual tamoxifen while maintaining HFD feeding, we performed metabolic phenotyping in these mice[29]. Gdf3[KO] mice had significantly decreased Gdf3 mRNA in their iWAT and eWAT, but not in skeletal muscle (Fig. 1B–D). There was no significant difference in body weight or body composition between groups (Fig. 1E and Supplementary Fig. 1B–D). We investigated the impact of Gdf3-deficiency on glucose homeostasis. Gdf3[KO] mice had significantly better glucose tolerance and insulin sensitivity than the control Gdf3[fl/fl] mice (Fig. 1F, G). Consistent with this result, circulating plasma insulin levels were significantly lower in the Gdf3[KO] mice after a 4 h fast (Fig. 1H).

As an endogenous ALK7 ligand[20], we examined whether GDF3 deficiency would affect free fatty acid (FFA) levels in obese mice. Under fasted conditions, we see a 40% reduction in basal circulating FFA in Gdf3[KO] mice. This decrease persists even after stimulation with isoproterenol, a non-selective agonist of β-adrenergic receptors (Fig. 1I). Similar effects were seen in HFD fed female Gdf3[KO] mice, but the differences were less pronounced, likely due to the fact that female C57Bl/6 J mice do not develop obesity and metabolic disease as readily as male mice (Supplementary Fig. 1E–H)[30]. All subsequent studies were performed in male mice. The analysis of inducible whole body Gdf3 KO mice suggests a phenotype of improved insulin sensitivity due to lower levels of adipose tissue lipolysis.

To further interrogate the importance of adipose tissue GDF3 in regulating lipolysis and insulin sensitivity, we also made eWAT-selective Gdf3 knockout mice (Gdf3[AKO]) by administering either control adeno-associated virus (AAV-GFP) or AAV-Cre to the eWAT of diet induced obese Gdf3[fl/fl] mice (Fig. 2A). We observed a decrease in Gdf3 mRNA in the eWAT of Gdf3[AKO] mice when compared to controls (Fig. 2B). We examined glucose tolerance in the Gdf3[AKO] mice using either a standard hand-held glucometer or a wireless continuous glucose monitoring (CGM) device. Similar to the whole-body inducible Gdf3[KO] mice, the fat-selective Gdf3[AKO] mice also had improved glucose tolerance with both measuring systems and reduced fasting plasma insulin levels without differences in body weight (Fig. 2C–G and Supplementary Fig. 2A). CGM also allowed us to examine glucose variability under unperturbed steady-state conditions. Blood glucose levels are tightly controlled in lean chow fed mice (Fig. 2H and ref. 31). Chronic HFD feeding leads to increased variability in blood glucose levels as seen in the HFD fed Gdf3[fl/f] controls. The Gdf3[AKO] mice however exhibited lower glucose variability when compared to the control group (Fig. 2H–J). Also similar to the whole-body inducible Gdf3[KO] mice, Gdf3[AKO] mice had lower basal and stimulated plasma FFA levels compared to controls (Fig. 2K). To investigate the effects of GDF3 deficiency on cold-induced lipolysis, we housed mice at 4 °C and found that Gdf3[AKO] mice had a significantly higher RER than Gdf3[fl/fl] mice (Fig. 2L–M). This increase in RER indicates that mice lacking GDF3 in their eWAT relied less on fat and more on carbohydrates as their energy source. This result is consistent with the phenotype of reduced adipose tissue lipolysis with less fat available for oxidation. These results indicate that both whole-body and epididymal adipose-selective deletion of Gdf3 improved glucose homeostasis, reduced circulating insulin levels, and reduced the use of fat as a substrate under thermogenic conditions. We next sought to understand the mechanism of how GDF3 loss of function in adipose tissue could drive whole-body insulin sensitivity.

### GDF3 loss of function decreases lipolysis in vitro

As rates of lipolysis integrate the signals from the central nervous system, catecholamine secretion and degradation, as well as intracellular signaling, we sought to examine the effect of Gdf3 deficiency under more controlled conditions, both ex vivo and in vitro. In ex vivo tissue explants from Gdf3[fl/fl] and Gdf3[KO] mice, we find basal FFA levels were lower in the iWAT explants of Gdf3[KO] mice (Fig. 3A). Upon stimulation of lipolysis with isoproterenol, both iWAT and eWAT explants from Gdf3[KO] mice had significantly lower FFA release

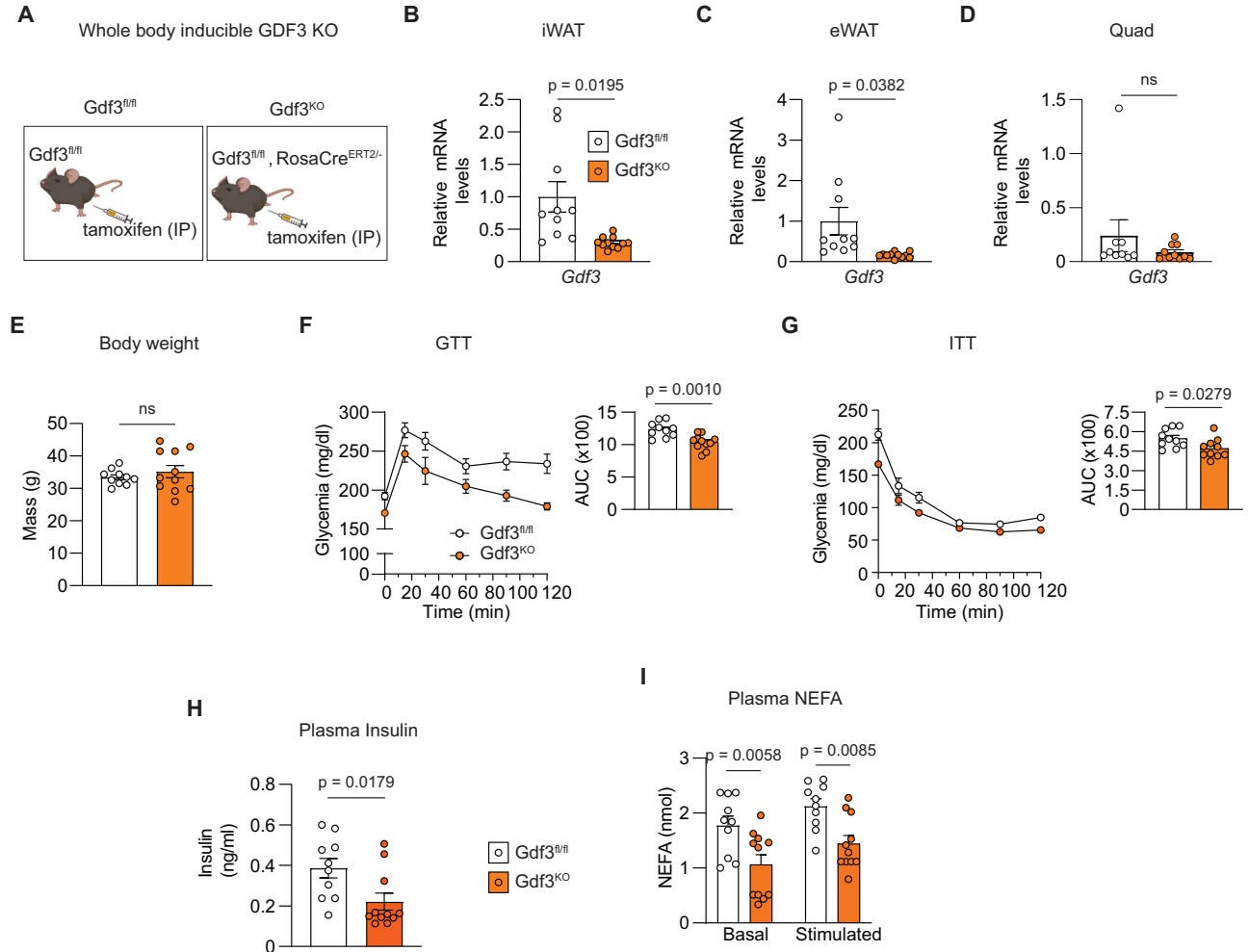

**Fig. 1 | Acute whole body GDF3 loss of function improves metabolic health in mice with diet-induced obesity. A** Graphical representation of adult male Gdf3^fl/fl and Gdf3^fl/fl::RosaCre^ERT2/- (Gdf3^KO) littermates fed a high-fat diet (HFD) for 8 weeks and then injected with tamoxifen intraperitoneally (IP) for 7 consecutive days. Created in BioRender. Banks, A. (2025) https://BioRender.com/to5r0d7. **B–D** RT-qPCR analysis of Gdf3 mRNA levels in white adipose tissue (WAT): inguinal (iWAT; $n = 10$ Gdf3^fl/fl, 11,Gdf3^KO), epididymal (eWAT; $n = 10$ Gdf3^fl/fl, 11 Gdf3^KO), and quadriceps skeletal muscle (Quad; $n = 9$ Gdf3^fl/fl, 10 Gdf3^KO) 9 weeks post tamoxifen. **E** Body weights during Glucose tolerance test (GTT, 4 weeks post tamoxifen; $n = 10$ Gdf3^fl/fl, 11 Gdf3^KO). **F** Left: GTT (4 weeks post tamoxifen; glucose dose: 1 g/kg BW,

IP). Right: Area under the curve (AUC) for GTT ($n = 10$ Gdf3^fl/fl, 11 Gdf3^KO). **G** Left: Insulin tolerance test (ITT) 5 weeks post tamoxifen (Insulin dose: 0.75U/kg BW, IP). Right: Area under the curve (AUC) for ITT ($n = 10$ Gdf3^fl/fl, 11 Gdf3^KO). **H** Fasting plasma insulin levels 6 weeks post tamoxifen ($n = 10$ Gdf3^fl/fl, 11 Gdf3^KO). **I** In vivo plasma non-esterified fatty acid (NEFA) levels, baseline and stimulated with isoproterenol for 15 min (10 mg/kg body weight) 6 weeks post tamoxifen ($n = 10$ Gdf3^fl/fl, 11 Gdf3^KO). **B–I** Data are presented as mean values +/− SEM. ns = not significant. Statistical comparisons were made using unpaired two-tailed Student's $t$-test (**B–H**) and two way ANOVA with Bonferroni's multiple comparisons test multiple comparisons (**I**). Source data are provided as a Source Data file.

compared to controls (Fig. 3A, B). While basal FFA levels did not change in the eWAT explants of Gdf3^AKO mice, the levels of FFA released from Gdf3^AKO mice was significantly lower upon isoproterenol stimulation (Fig. 3C). There were no differences in either basal or stimulated FFA release from the iWAT explants of Gdf3^AKO mice (Fig. 3D), consistent with the localized deletion of Gdf3 from the eWAT of these mice.

We next examined lipolysis in vitro in primary adipocytes with acute Gdf3 deficiency. We isolated SVF from the iWAT of Gdf3^fl/fl mice with or without Rosa-Cre^ERT2 and fully differentiated them into primary adipocytes in vitro before administering 4-hydroxytamoxifen (4-OHT) for acute deletion of Gdf3. In this purified population of adipocytes, Gdf3 mRNA levels were reduced by 80% in the Cre expressing Gdf3^KO cells (Fig. 3E). We observed no significant differences in adipocyte differentiation markers adiponectin (*Adipoq*), *PPARγ*, *Leptin*, and pre-adipocyte factor 1 (*Pref1*) between the two groups of cells (Supplementary Fig. 2B). There was also no difference in the amount of oil red O incorporated into the two groups of cells (Supplementary Fig. 2C).

The loss of Gdf3 in primary adipocytes resulted in no change in basal levels but significantly reduced FFA release in response to β-adrenergic activation with isoproterenol for 2 h (Fig. 3F). Our results show that loss of function of Gdf3 led to reduced basal and stimulated levels of plasma FFA in vivo and reduced free fatty acid release in response to noradrenergic stimulation both ex vivo, in tissue explants, and in vitro, in differentiated primary adipocytes. We next examined the components of lipolytic signaling. Despite similar levels of adipocyte differentiation, we observed decreased phosphorylation of targets of protein kinase A (pPKA substrates) as well as reduced expression of β3 adrenergic receptor (β3-AR), and lower levels of phosphorylated hormone sensitive lipase (pHSL), in the Gdf3 deficient cells when stimulated with isoproterenol for increasing amounts of time (Fig. 3G, H and Supplementary Fig. 2D, E). RT-qPCR analysis of *Adrb3* mRNA from primary adipocytes cultured from iWAT SVF cells of Gdf3^fl/fl and Gdf3^KO mice also showed significantly lower expression levels within 5 min of isoproterenol stimulation that persisted over time (Fig. 3I). This was accompanied by lower cAMP levels in primary adipocytes derived

from Gdf3[KO] mice at 5 and 15 min following isoproterenol stimulation (Fig. 3J). When looking at time-dependent FFA release in these cells, Gdf3[KO] primary adipocytes showed significantly lower FFA release within 5 min of isoproterenol stimulation and this difference increased in a time dependent manner. In fact, the addition of the cAMP analog (8-Bromo-cAMP) was able to completely restore the ability of GDF3[KO] primary adipocytes to release FFA (Fig. 3K). Confocal imaging of primary adipocytes also reconfirmed these effects, with adipocytes from Gdf3[KO] mice having visibly lower β3-AR and cAMP levels, 5 and 15 min after isoproterenol stimulation respectively (Fig. 3L, M). These effects are consistent with decreased cellular lipolytic signaling in cells lacking GDF3.

## Effect of GDF3 and TGFβ superfamily ligands on SMAD signaling

The effects of GDF3 as a regulator of lipolysis and glucose homeostasis are better understood in the context of the TGFβ superfamily. Different members of the TGFβ superfamily may have similar or opposing functions. BMPs, TGFβ and activin-like ligands usually bind to heteromeric receptors and either phosphorylate SMAD 1, 5, and 9 to activate BMP signaling in the case of BMP ligands or phosphorylate SMAD2 and SMAD3 to activate TGFβ/activin-like signaling. Both TGFβ and activins can also inhibit SMAD 1,5,9 signaling[5]. These phosphorylated SMADs can interact with various transcriptional activators, co-activators, and co-repressors to regulate gene expression. GDF3 has been reported to act like a classical BMP antagonist similar to Noggin

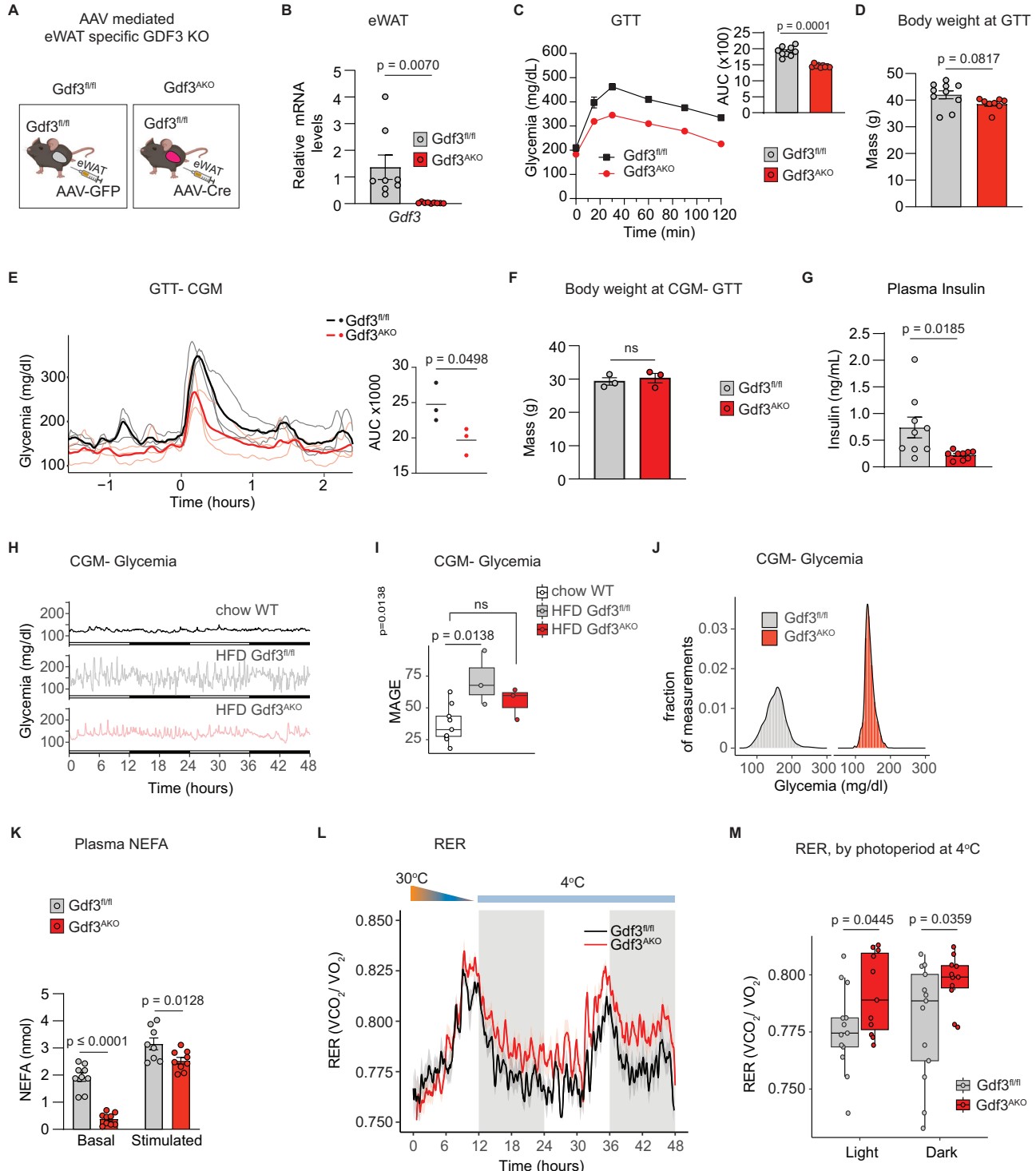

**Fig. 2 | Acute GDF3 loss of function in epididymal adipose tissue improves metabolic health in mice with diet-induced obesity. A** Graphical representation of adult male Gdf3[fl/fl] HFD fed mice administered AAV-eGFP (Gdf3[fl/fl]) or AAV-Cre (Gdf3[AKO]) into eWAT. Created in BioRender. Banks, A. (2025) https://BioRender.com/to5r0d7. **B** RT-qPCR analysis of *Gdf3* in eWAT, 6 weeks post AAV administration. **C** Left: GTT, 2 weeks post AAV administration (glucose dose: 1 g/kg BW, IP). Right: Area under the curve (AUC) for GTT.D) Body weights during GTT in (**C**). **B**–**D** *n* = 8 Gdf3[fl/fl], 9 Gdf3[AKO]. **E**–**J** Continuous glucose monitoring (CGM) of HFD fed Gdf3[fl/fl], Gdf3[AKO] male mice. **E** Left: GTT-CGM, 4 weeks post AAV injections, (glucose dose: 1 g/kg BW, IP). Right: AUC of GTT-CGM, (*n* = 3). **F** Body weights during GTT-CGM (*n* = 3). **G** Fasting plasma insulin (*n* = 8 Gdf3[fl/fl], 9 Gdf3[AKO]). **H** CGM in ad libitum chow fed WT mice (*n* = 8), HFD fed Gdf3[fl/fl] and Gdf3[AKO] mice (*n* = 3). **I** Mean Amplitude of Glycemic Excursion (MAGE), representing glycemic variability from data in (**H**) (*n* = 8 WT, 3 Gdf3[fl/fl] and 3 Gdf3[AKO]). **J** Histograms representing relative frequency of glycemia measurements over 24 h in a single ad libitum fed Gdf3[fl/fl] or Gdf3[AKO] mouse. **K** In vivo plasma NEFA levels, baseline and stimulated with isoproterenol for 15 min (*n* = 8 Gdf3[fl/fl], 9 Gdf3[AKO]). **L** Respiratory exchange ratio (RER), 2 weeks post AAV injections during cold exposure(*n* = 11 Gdf3[fl/fl], 13 Gdf3[AKO]). **M** Box plots of RER at 4 °C of Gdf3[fl/fl] and Gdf3[AKO] from (**L**) averaged over light and dark photoperiods (*n* = 11 Gdf3[fl/fl], 13 Gdf3[AKO]). **B**–**G**, **K**, **L** Data are presented as mean values +/− SEM, ns = not significant. **I**, **M** Box plots presenting the median as the center, 25% and 75% percentiles as box limits and whiskers extending to the largest and smallest values within the 1.5 × interquartile range of the box limits. Statistical comparisons were made using unpaired two-tailed Student's *t*-test (**B**–**G**), or one way ANOVA (**I**) or unpaired two-tailed Welch's unequal variances *t*-test (**M**) or two way ANOVA with Sidak's multiple comparisons test (**K**). Source data are provided as a Source Data file.

or Gremlin to block SMAD1/5/8 activation[32,33]. GDF3 is also reported to activate SMAD2/3 signaling through the ALK7 receptor[20]. The effects of BMPs or activin/TGFβ signaling on lipolysis are varied. We examined the expression changes of genes that respond to BMP proteins[34] or TGFβ[35] in adipose tissue in the presence or absence of GDF3. Gene expression in eWAT and iWAT of GDF3[KO] mice revealed increased expression of genes downstream of BMP signaling: *Id1, Id2, Id4, Hdac4, Lamba3, Zbtblb, Snai2* and *Slpi* (Fig. 4A and Supplementary Fig. 2F). Gdf3[KO] mice simultaneously showed reduced expression of TGFβ signaling target genes including *Adamts4, Adamts5, Adamts7, Col3a1, Col5a3, Col18a1*, and *Col27a1* in their iWAT and/or eWAT (Fig. 4B and Supplementary Fig. 2G). Similar gene expression changes were also seen in the eWAT of Gdf3[AKO] mice (Supplementary Fig. 2H, I). These data support the hypothesis that GDF3 regulates BMP and TGFβ signaling in adipose tissues of adult animals.

To further examine the effects of GDF3 on BMP and TGFβ/activin like signaling, we generated a dual fluorescent reporter system using BRE-YFP and SBE-RFP reporters in immortalized iWAT preadipocytes to analyze SMAD signaling using flow cytometry (Fig. 4C, D and[36–38]). These cells express most of the relevant type 1 and type 2 receptors as well as the coreceptors commonly involved in BMP and TGFβ/activin like signaling, based on the previously published RNA seq dataset from these cells (Supplementary Fig. 3A)[39]. Under serum-free conditions recombinant human BMP9 (rhBMP9) led to strong BMP signaling dependent YFP expression in a dose dependent manner, but did not affect the SBE-RFP reporter in these cells. Both recombinant mouse GDF3 (rmGDF3) and rhTGFβ1, in contrast, inhibited the BRE-YFP reporter, and led to dose dependent increased expression of the SBE-RFP reporter (Fig. 4E, F). This system helped us examine the impact of TGFβ superfamily ligands on both BMP and TGFβ/activin- like signaling reporters simultaneously in the same cells. This reporter system faithfully recapitulated the reciprocal effect of GDF3 on gene expression in vivo.

## GDF3 is a BMP signaling antagonist with a high affinity for the receptor BMPRII

To further explore GDF3's role in BMP signaling in the presence of BMP ligands we tested the effect of rmGDF3 on rhBMP2, rmBMP7, rhBMP9, and rmBMP10 in serum-free conditions in the dual reporter immortalized iWAT preadipocytes. Indeed, rmGDF3 was able to inhibit all recombinant BMPs tested suggesting that GDF3 is a broad inhibitor of BMP signaling (Fig. 5A).

Next, we conducted ligand-receptor interaction assays to investigate whether GDF3 acts as a pan BMP inhibitor through interaction with BMP signaling receptors. Using Surface Plasmon Resonance (SPR) analysis, we found that rhGDF3 has a high affinity for the Fc-chimera of the type II BMP receptor (rhBMPR2-Fc) (Fig. 5B and Table 1). This interaction was also confirmed through biolayer interference (BLI) analysis, which showed that both human and mouse rGDF3 can bind to rhBMPR2-Fc with high affinity (Fig. 5C, D and Table 1). Additionally, isothermal titration calorimetry (ITC) analysis with rmGDF3 against rhBMPR2-Fc again confirmed the interaction between this ligand and receptor combination (Fig. 5E and Table 1). All three assays demonstrated binding interaction between GDF3 and BMPR2 with binding constants reported in Table 1. We saw negligible interaction between rmGDF3 and BMP ligands- rhBMP2, rmBMP4 and rhBMP9 in an ITC assay (Supplementary Fig. 3B and Table 1).

Noting previous concerns about TGFβ contamination in purified recombinant proteins derived from CHO cells[40], we aimed to rule out the possibility that trace amounts of TGFβ protein in our recombinant GDF3 preparations were causing our observed effect of BMP signaling inhibition. We created vectors with the Tet-On inducible gene expression system to induce the expression of either a control blue fluorescent protein (BFP) referred to as TetON-Control, or full-length mouse GDF3 and BFP (TetOn-mGdf3) in the presence of the tetracycline analog doxycycline, giving us temporal control over the expression of these proteins. Western blot analysis of 293T cells transfected with the TetOn-Control or TetOn-mGdf3 vectors showed strong full-length GDF3 protein expression only in TetOn-mGdf3 expressing cells and only in the presence of doxycycline (Fig. 5F). We next transfected these vectors encoding either the control or Gdf3 gene into C2C12 myoblasts, containing the same BRE-YFP and SBE-RFP reporters stably integrated into them. We treated these cells with increasing doses of doxycycline in serum-free conditions for 48 h. The TetOn-mGdf3 expressing cells showed a significant dose-dependent increase in Gdf3 mRNA upon doxycycline treatment (Fig. 5G). We then tested whether genetically encoded Gdf3 could activate the BRE-YFP reporter. Doxycycline induction, in the absence of BMP ligands, did not affect the BRE-YFP reporter activity in the TetOn-control or TetOn-mGdf3 expressing cells (Supplementary Fig. 2C, D). However, doxycycline induced GDF3 attenuated the reporter's response to increasing doses of rhBMP2 (green bars) compared to the control cells (yellow bars) (Fig. 5H). These results are consistent with our reporter assays where rmGDF3 inhibits BMPs. These data suggest that GDF3 blocks BMP signaling by acting as a BMPR2 antagonist.

## GDF3 acts as a TGFβ/activin-like SMAD2/3 signaling agonist with a high affinity for receptors ACVR2A and ACVR2B

To further confirm the autonomous ability of GDF3 to activate the SBE-RFP reporter (as shown in Fig. 4F), we performed ligand-receptor interaction studies to determine the receptors involved in GDF3's ability to activate SMAD2/3 signaling. SPR analysis of rhGDF3 with chimeric rhACVR2A-Fc, rhACVR2B-Fc, and rhTGFBR2-Fc receptors showed that GDF3 had a high affinity for ACVR2A and ACVR2B but had no binding affinity for TGFBR2 (Fig. 6A and Table 1). High affinity for ACVR2B is consistent with previous results[41].

Next, we analyzed if GDF3 competes with other ACVR2A/2B-activating ligands. We added rmGDF3 with rhTGFβ1 in serum-free conditions to our dual reporter iWAT preadipocytes. Recombinant mGDF3 had synergistic effects on the SBE-RFP reporter activity with rhTGFβ1,

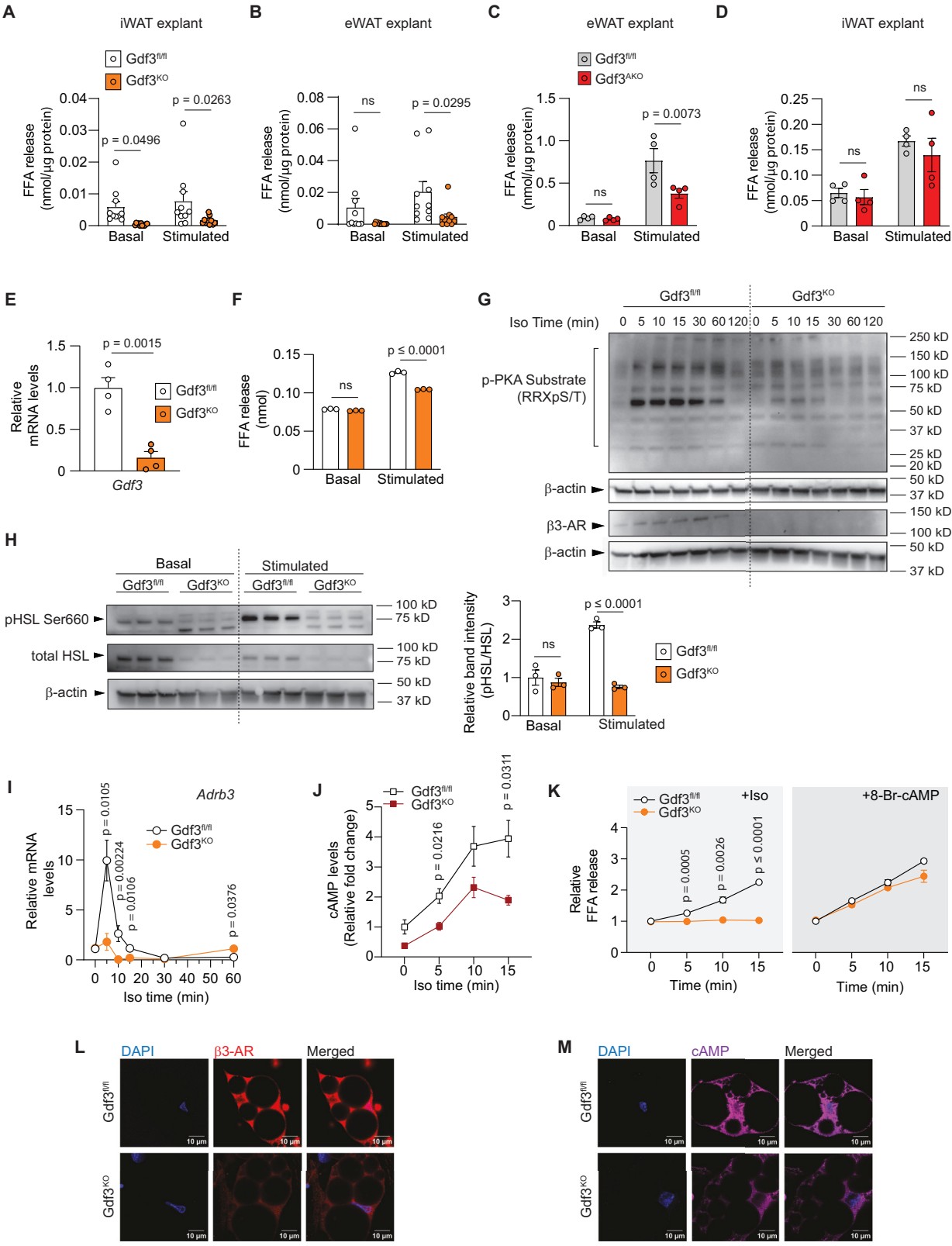

leading to significantly higher reporter activity than either ligand alone (Fig. 6B). These results suggest that GDF3 acts as a SMAD2/3 signaling ligand in the absence of serum and in the presence of TGFβ1.

To address the possibility of trace levels of contaminating TGFβ in CHO cell-derived recombinant protein samples, we used our genetically encoded TetOn-mGdf3 or control vectors to perform reporter assays in the presence of doxycycline induced Gdf3 expression in

C2C12 myoblasts. The result showed that doxycycline induced GDF3 could activate the SBE-RFP reporter in a dose-dependent manner (Fig. 6C). Cells transfected with the TetOn-control vector did not show any SBE-RFP reporter activity in response to increasing doses of doxycycline (Supplementary Fig. 2E).

Lastly, we examined the relative activation of type 1 receptors ALK4/5/7 by rmGDF3. In conventional SBE-luciferase assays,

**Fig. 3 | GDF3 loss of function leads to decreased lipolysis. A**, **B** Free fatty acid (FFA) release from iWAT and eWAT explants of Gdf3^fl/fl and Gdf3^KO mice on HFD (from Fig. 1) at baseline and following isoproterenol stimulation ($n = 10$ Gdf3^fl/fl, 11 Gdf3^KO). **C**, **D** FFA release from iWAT and eWAT explants of Gdf3^fl/fl and Gdf3^AKO mice (from Fig. 2) at baseline and following isoproterenol ($n = 4$). **E**–**M** Primary adipocytes from differentiated stromal vascular fractions (SVF) of iWAT of Gdf3^fl/fl and Gdf3^fl/fl::RosaCre^ERT2/− mice, treated with 4-hydroxytamoxifen (4-OHT) to generate Gdf3^fl/fl and Gdf3^KO cells. **E** Gdf3 mRNA levels ($n = 4$). **F** FFA release at baseline and following isoproterenol stimulation for 2 h ($n = 3$). **G**, **H** Western blot analysis following isoproterenol stimulation. **G** Protein expression of phosphorylated protein kinase A (p-PKA) substrates, β3- adrenergic receptor 3 (β3-AR) and β-actin (loading control) ($n = 1$ per time point). **H** Left: Protein expression of phosphorylated hormone-sensitive lipase at serine 660 (pHSL Ser 660) corresponding total HSL and β-actin levels without (basal) and with isoproterenol (stimulated) for 15 min. Right: Relative band intensities of pHSL versus total HSL quatified from the left panel, ($n = 3$). **I** RTqPCR analysis of Adrb3 in response to isoproterenol stimulation ($n = 6$). **J** Relative cAMP levels in response to isoproterenol stimulation ($n = 3$). **K** Relative FFA release in response to isoproterenol or 8-Bromo-cAMP at indicated time points ($n = 3$). **L** Immunofluorescence (IF) staining of β3-AR (red), 5 min post isoproterenol stimulation. Nuclear stain: DAPI (blue). **M** IF staining of cyclic AMP (cAMP) (purple), 15 min post isoproterenol stimulation. Nuclear stain: DAPI (blue). **A**–**F**, **H**–**K**) Data are presented as mean values +/− SEM, ns = not significant. Statistical comparisons were made using two way ANOVA with Sidak's multiple comparisons test (**A**–**D**, **F**, **H**) or unpaired two-tailed Student's *t*-test (**E**, **I**–**K**) for each time point. Source data are provided as a Source Data file.

HEK293T cells were initially treated with increasing doses of rmGDF3 along with an empty vector (EV), which led to strong dose dependent increase in luciferase reporter activity as expected. In cells treated with SB-431542 (inhibitor of type 1 receptors ALK4/5/7)[42], rmGDF3 lost all ability to stimulate SBE reporter activity via SMAD2/3 (Fig. 6D). When these cells were made to express SB-resistant ALK4, ALK5, or ALK7 receptors by transient transfection (ALK4/5/7-ST) to restore activation of the specific endogenous type 1 receptor signaling, rmGDF3 could maximally increase reporter activation through SB-resistant ALK5 by 70-fold while SB-resistant ALK4 had minimal effects and some activation by SB-resistant ALK7 (Fig. 6D). These data suggest that ALK5 and ACVR2A/2B may be prominent receptors in mediating GDF3's ability to activate SMAD2/3 signaling, but does not exclude the possibility of cell type specific effects or the requirement of co-receptors.

## Recombinant GDF3 enhances lipolysis in vitro in cultured mature adipocytes

To understand the effect of TGFβ superfamily proteins on lipolysis, we treated cultured mature adipocytes (differentiated from immortalized murine iWAT preadipocytes) with recombinant proteins. We found that rmGDF3, like rhTGFβ1, leads to a dose-dependent release of free fatty acids in response to isoproterenol (Fig. 7A). In contrast rhBMP2, rhBMP9, and rmBMP10 all significantly reduced isoproterenol stimulated free fatty acid release (Fig. 7B). These effects were independent of any changes in adipocyte differentiation (Supplementary Fig. 3F–H). We next examined whether these effects were dependent on adipose triglyceride lipase (ATGL), a critical enzyme required for lipolysis. Cultured cells treated with Atglistatin, an ATGL inhibitor[43], had approximately 50% reduced levels of stimulated lipolysis compared to cells without the inhibitor. However, even with reduced rates of lipolysis, rmGDF3 and rhTGFβ1 both increased lipolysis while rhBMP9 decreased lipolysis (Fig. 7C). These effects suggest that the actions of GDF3 to activate lipolysis are not mediated through ATGL.

Considerable crosstalk exists between BMP and TGFβ/activin like signaling where activins can impact BMP signaling and vice versa. To understand whether GDF's effects on lipolysis occur through TGFβ/activin or BMP signaling or both, we treated cultured mature adipocytes with inhibitors of either pathway. In the absence of inhibitors as before, both rmGDF3 and rhTGFβ1 activated isoproterenol stimulated lipolysis while rhBMP9, rhBMP2 and rmBMP4 inhibited lipolysis (Fig. 7D and Supplementary Fig. 3I, J). We used LDN-193189 to inhibit BMP signaling through the type 1 BMP receptors (ALK1/2/3/6)[44]. We observed no impact of inhibiting BMP signaling on rmGDF3 or rhTGFβ induced lipolysis. However, rhBMP9 as well as rhBMP2 and rmBMP4 could no longer inhibit FFA release (Fig. 7D and Supplementary Fig. 3I, J). Next, we examined the dependence on the activin/TGFβ type 1 receptors (ALK4/5/7) using the SB-431542 inhibitor. In the presence of the TGFβ/activing like signaling inhibitor, neither rmGDF3 nor rhTGFβ1 increased lipolysis while all three rBMPs retained the ability to inhibit lipolysis (Fig. 7D and Supplementary Fig. 3I, J). These results suggest that GDF3's ability to stimulate lipolysis requires signaling through the activin/TGFβ type 1 receptors. The findings of rmGDF3 promoting lipolysis are consistent with the inverse observation wherein GDF3 deficiency is associated with decreased lipolysis. To further delineate how GDF3 enhances lipolysis, we examined the components of lipolytic signaling. We observed increased phosphorylation of targets of protein kinase A (pPKA substrates) in a time dependent manner with isoproterenol stimulation. We did not see clear differences in the protein levels of β3-AR in the cells treated with rmGDF3 and isoproterenol for increasing amounts of time in a western blot analysis, but did see that cells pretreated with rmGDF3 showed significantly higher pHSL levels after 15 min of isoproterenol stimulation (Fig. 7E, F). We did interestingly see an increase in Adrb3 mRNA levels in cells treated with rmGDF3 with Adrb3 levels peaking 10 min post isoproterenol treatment (Fig. 7G). Using confocal imaging following immunofluorescence staining we also saw higher levels of β3-AR in cells pretreated with rmGDF3 and stimulated with isoproterenol for 5 min (Fig. 7H). This was accompanied by increased cAMP levels following isoproterenol treatment in cells preexposed to rmGDF3 for the indicated time points (Fig. 7I, J). In the presence of the cAMP analog, 8-Bromo-cAMP, both control and rmGDF3 treated cells showed similar increases in FFA release, unlike isoproterenol stimulation where rmGDF3 treated cells showed increased FFA release (Fig. 7K). These results corroborate our loss of function GDF3 results and suggest that GDF3's effect on enhancing lipolysis occurs upstream of the release of cAMP and involves the transcriptional regulation of Adrb3.

## Discussion

Adipose tissue holds most of the body's energy reserves, and environmental and physiological cues tightly regulate rates of storage and release. In obesity, adipose tissue undergoes increased rates of lipolysis contributing to ectopic lipid accumulation[45]. Our data suggest that GDF3 works on adipocytes to increase beta-adrenergic signaling and increase rates of lipolysis. While TGFβ superfamily proteins affect mammalian glucose and energy metabolism, much of our understanding of their effects in adipose tissue is ascribed to roles in cellular differentiation, thermogenesis, and fibrosis. A useful, if oversimplified framework suggests that increased BMP signaling has salutary effects to improve insulin resistance[46,47] while in contrast, elevated TGFβ1 signaling has adverse effects in promoting tissue fibrosis and inflammation[48–50]. Activins and GDF proteins may use receptors in both pathways. To understand the biology of GDF3, we developed a tool to simultaneously assess BMP receptor and TGFβ and activin receptor signaling.

BMP receptor signaling leads to phosphorylation of SMAD1, SMAD5, and/or SMAD8. Similarly, TGFβ and activin signaling induce the phosphorylation of SMAD2, and SMAD3. Immunoblotting for protein phosphorylation has served as the backbone for understanding ligand and receptor signaling[51]. Luminescence reporter assays for BMP signaling have used the Id1 promoter's BMP-Responsive Element (BRE)[52]. Likewise, TGFβ signaling has used luminescence

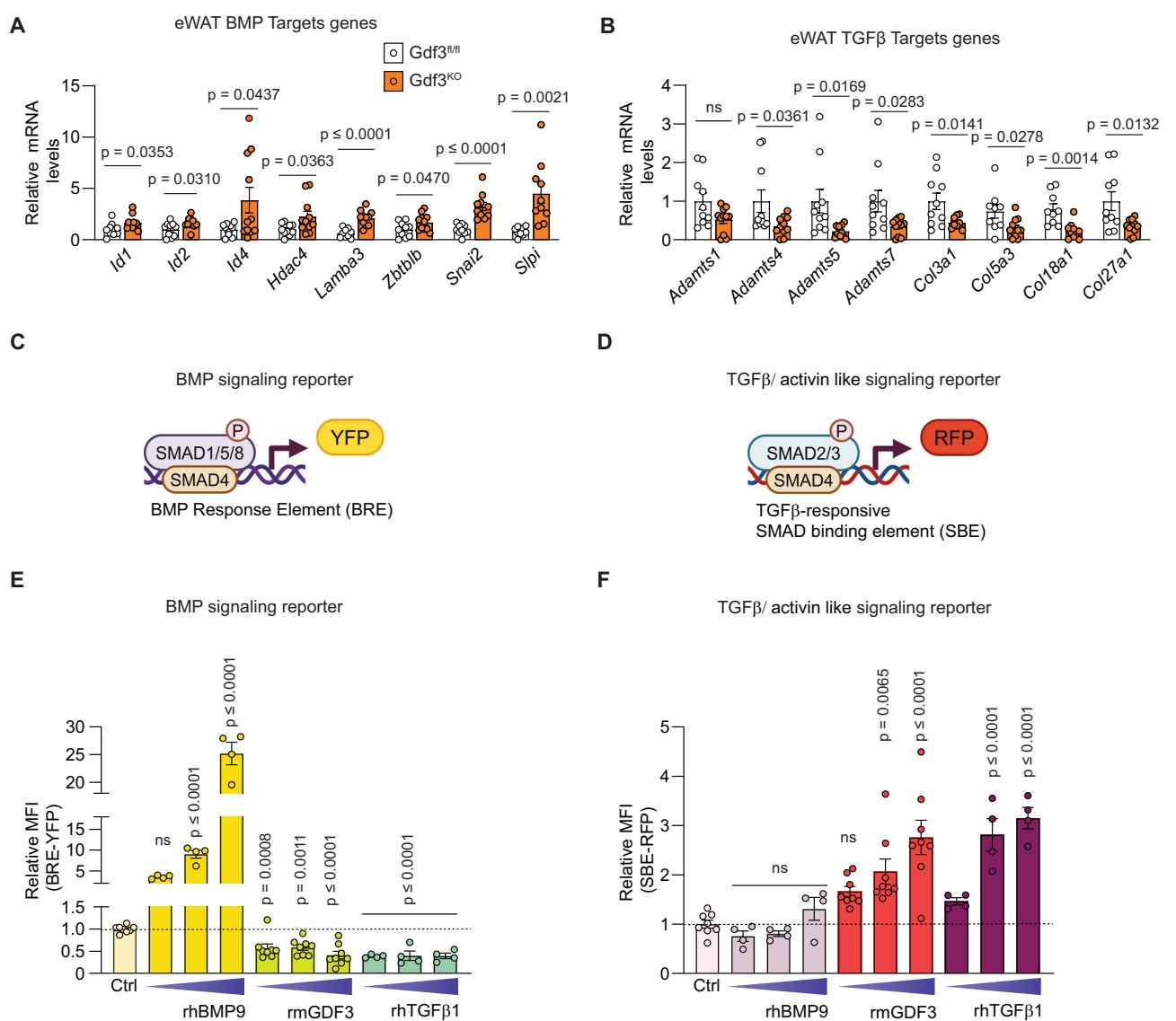

**Fig. 4 | GDF3 regulates SMAD signaling. A-B** RT-qPCR analysis of genes in eWAT of Gdf3$^{fl/fl}$ and Gdf3$^{KO}$ mice from Fig. 1. **A** BMP signaling targets (*n* = 10 Gdf3$^{fl/fl}$, 11 Gdf3$^{KO}$ for each gene, with the exception of *Lamba3* and *Slpi* (*n* = 9 for Gdf3$^{fl/fl}$) and *Snai2* and *Slpi* (*n* = 10 for Gdf3$^{KO}$). **B** TGFβ signaling targets (*n* = 10 Gdf3$^{fl/fl}$, 11 Gdf3$^{KO}$ for each gene, with the exception of *Col5a3* and *Col18a1* (*n* = 9 Gdf3$^{fl/fl}$) and *Col18a1* (*n* = 10 Gdf3$^{KO}$). **C, D** Graphical representation of the BMP responsive Smad1/5/8 dependent BRE-YFP reporter (**C**) and the TGFβ/ activin responsive Smad2/3 dependent SBE-RFP reporter (**D**) created in BioRender. Banks, A. (2025) https://BioRender.com/bzkblq8 (**C**) and Banks, A. (2025) https://BioRender.com/k1zb4oy (**D**). **E, F** Flow cytometry analysis of dual reporter iWAT preadipocytes incubated with increasing doses of recombinant mouse (rm)GDF3, recombinant human (rh) TGFβ1 and rhBMP9 in serum free conditions. Relative Mean Fluorescence Intensity (MFI) of BRE-YFP (**E**) or SBE-RFP (**F**). Doses: rmGDF3 (250, 500 and 1000 ng/mL), rhTGFβ1 (2.5, 5 and 10 ng/mL) and rhBMP9 (63, 125, 250 ng/mL), (n = 8 for Ctrl and each dose of rmGDF3, *n* = 4 for each dose of rhTGFβ1 and rhBMP9, biological replicates, where each n represents the average relative MFI of approximately 1000 sorted live cells). **A, B, E, F** Data are presented as mean values +/− SEM, ns = not significant. Statistical comparisons were made using unpaired two-tailed Student's *t*-tests for each gene (**A, B**) and one way ANOVA with Dunnet's multiple comparison test for each recombinant compared to the Ctrl group (**E, F**). Source data are provided as a Source Data file.

driven by the SMAD3-Binding Element (SBE or CAGA$_{12}$)[53]. The Elowitz group at Caltech recently created a flow cytometry-compatible BMP reporter with the BRE promoter driving expression of Citrine, a yellow fluorescent protein (BRE-YFP)[36–38]. This has further increased throughput and allowed for a systems-level approach to BMP signaling not previously possible. We extended these studies to generate a red fluorescent TGFβ signaling reporter driven by the SBE promoter (SBE-RFP). Using the readouts of both reporters is likely to help unravel the complex crosstalk of TGFβ superfamily ligand signaling.

Surprisingly, we find the loss of GDF3 alters gene expression to both decrease TGFβ signaling and increase BMP target gene

expression in vivo. These effects are independent of cellular differentiation (Supplementary Fig. 2). Our in vitro findings with the dual fluorescent reporter have confirmed this regulation with both recombinant GDF3 and with an inducible plasmid-encoded GDF3 in preadipocytes, or myoblasts[54].

The present study shows that GDF3 may inhibit BMP signaling by acting as a type II receptor antagonist, an alternative explanation to the previously accepted model where GDF3 directly interacts with BMP ligands[3]. Our study establishes that GDF3 has a high affinity for multiple type II receptors: BMPR2, ACVR2A, and ACVR2B but not TGFBR2. These interactions with multiple type II receptors are

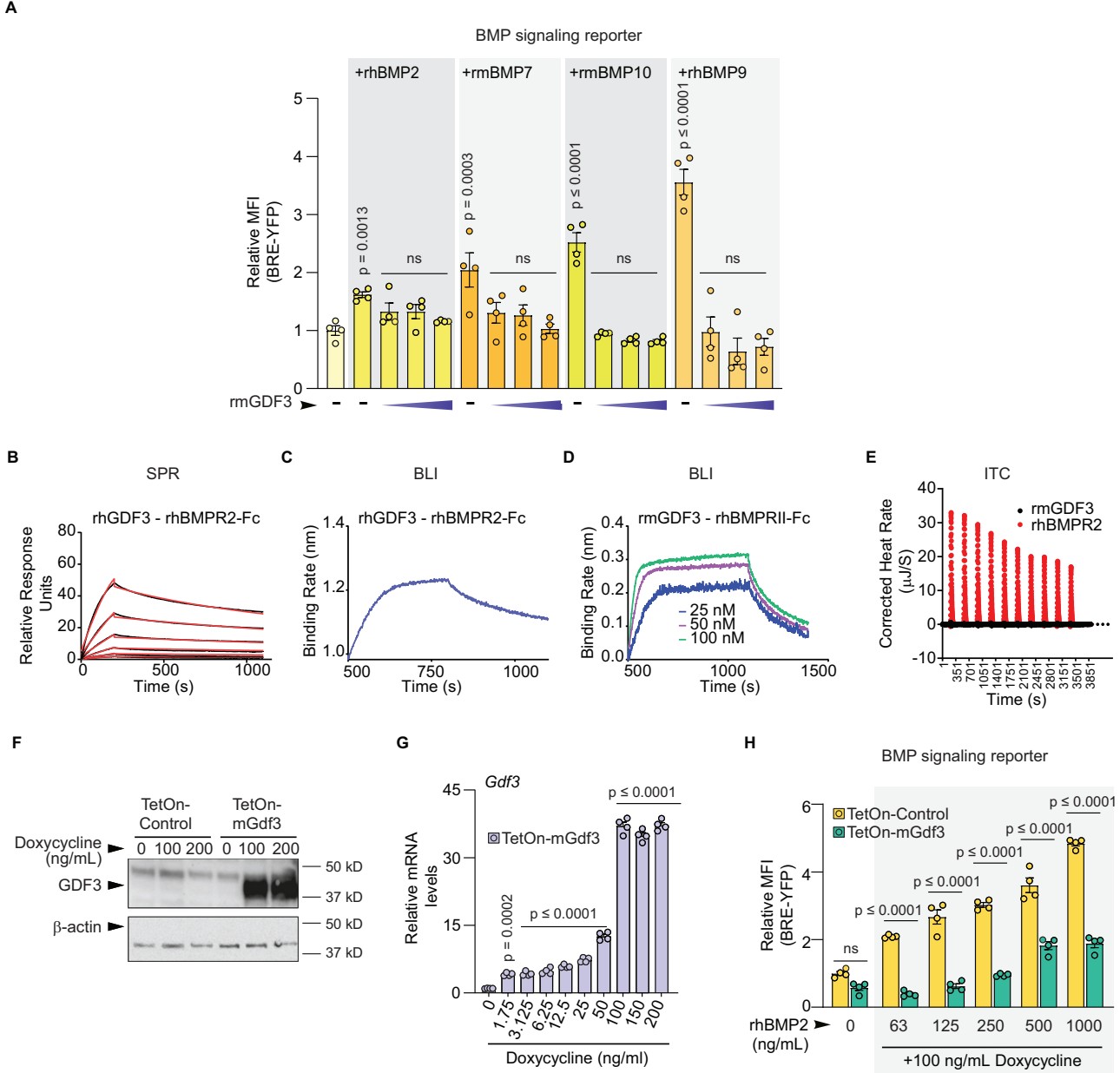

**Fig. 5 | GDF3 is a BMP signaling antagonist with a high affinity for the receptor BMPRII. A** Relative MFI of BRE-YFP in dual reporter preadipocytes treated with serum free media (control) or 63 ng/mL rhBMP2, rmBMP7, rmBMP10, or rhBMP9 alone or with increasing doses of rmGDF3 at 250, 500 and 1000 ng/mL (*n* = 4 biological replicates for each group. Each *n* represents the average relative MFI of approximately 1000 sorted live cells). **B** Surface Plasmon Resonance (SPR) analysis of rhGDF3 with rhBMPR2-Fc. The raw binding curves are in black, and the kinetic binding fits are in red. **C** Biolayer Interferometry (BLI) analysis of 50 nM rhGDF3 interacting with rhBMPR2-Fc. **D** BLI analysis of 25–100 nM of rmGDF3 with rhBMPRII-Fc. **E** Isothermal Titration Calorimetry (ITC) assay showing the binding isotherm for the titration of 250 ng/mL rhBMPRII-Fc against 250 ng/mL of rmGDF3. **F–H** Cultured cells transiently transfected with doxycycline-inducible TetOn-Control or TetOn-mGdf3 vectors for 24 h and incubated with doxycycline for 48 h.

**F** Western blot analysis of HEK293T cells. Protein expression levels of GDF3 and β-actin (loading control). **G, H** C2C12 dual reporter myoblasts All cells were harvested for RNA isolation (**G**), or cells expressing YFP, RFP, and BFP were sorted and analyzed for changes in reporter activity (**H**). **G** RTqPCR analysis of *Gdf3* (*n* = 4 biological replicates, where each n represents all the cells from one well of a 12 well plate). **H** Relative MFI of BRE-YFP in sorted YFP, RFP, and BFP triple positive cells treated with doxycycline and rhBMP2 (*n* = 4 biological replicates per group where each *n* represents the average relative MFI of all the triple positive live cells per well). **A, G, H** Data are presented as mean values +/− SEM, ns = not significant. Statistical comparisons were made using one way ANOVA with Dunnet's multiple comparison test (**A, G**) or two way ANOVA with Tukey's multiple comparisons test (**H**). Source data are provided as a Source Data file.

analogous to activin A and B[55] and suggest that GDF3 plays a biological role as an activin-like signaling molecule. According to our studies, GDF3 can broadly inhibit BMP activity. Loss of function of Gdf3 increases mRNA expression of endogenous BMP-stimulated genes. GDF3 can inhibit the reporter activity of several BMPs, indicating that it could inhibit BMP signaling through one of the BMPR2, ACVR2A, and ACVR2B receptors[55]. The increased expression of GDF3 observed in obesity may contribute to pathologies linked to "BMP resistance"−where despite increasing circulating levels of ligands, BMP signaling is diminished[56,57]. These findings are consistent with crosstalk where reduced formation of BMP receptor signaling complexes enhances SMAD2/3 signaling[55,58].

**Table. 1 | Kinetics data from protein - protein interaction studies**

| Analysis | Ligand1 | Receptor/ Antibody/ Ligand2 | ka (1/Ms) | kd (1/s) | KD (M) |
|---|---|---|---|---|---|
| SPR | rhGDF3 | rhBMPR2 | 2.19E + 06 | 0.001493 | 6.01E-10 |
| SPR | rhGDF3 | rhACVR2A | 1.21E + 06 | 4.05E-04 | 1.26E-09 |
| SPR | rhGDF3 | rhACVR2B | 2.12E + 06 | 3.80E-04 | 3.32E-10 |
| SPR | rhGDF3 | rhTGFBR2 | N/A | N/A | N/A |
| BLI | rmGDF3 | rhBMPR2 | 3.703E + 05 | 4.833E-03 | 1.310E-08 |
| ITC | rmGDF3 | rhBMPR2 | 1.233E + 06 | | 8.111E-07 |
| ITC | rmGDF3 | rhBMP2 | 1.00E + 09 | | 1.00E-09 |
| ITC | rmGDF3 | rmBMP4 | 1.00E + 09 | | 1.00E-09 |
| ITC | rmGDF3 | rhBMP9 | 1.00E + 09 | | 1.00E-09 |

GDF3 can activate activin-like reporter activity and the expression of TGFβ-induced genes both in vitro and in vivo. This occurs because GDF3 has a high affinity for the type II receptors ACVR2A and ACVR2B. This finding is consistent with a previously reported co-immunoprecipitation study of ACVR2B and GDF3[5]. ALK7 in the presence of the co-receptor cripto has been reported to serve as the type I receptor responsible for GDF3 signaling in adipose tissue[20]. Our in vitro data did not include cripto in the assay which could explain the modest reporter response to ALK7, but suggests that GDF3 may also activate ALK5 to mediate intracellular signaling. Both ALK5 and ALK7 are expressed in adult adipose tissues, while cripto is not[59,60].

The results of Gdf3 deficiency to lower levels of lipolysis are unexpected if we assume that GDF3 acts as an agonist ligand of the ALK7 receptor. Acute adipose tissue deletion of ALK7 (Alk7fl/fl::Adipoq-CreERT2) in high fat-fed mice has minimal effects on body weight or lipolysis under steady-state HFD conditions[61]. However, treating obese mice with an ALK7-neutralizing antibody or ALK7 siRNA knockdown consistently reduces body weight in obese mice[24,25]. These treatments also produce increased rates of lipolysis and concomitant insulin resistance despite lower fat mass. In contrast, constitutively active ALK7 decreases lipolysis[62]. From these studies, ALK7-driven signaling acts as a brake on lipolysis, while decreasing ALK7 signaling or expression liberates this braking mechanism. These conclusions are supported by regulation of adiposity by activin E, an ALK7 agonist ligand[63]. Activin E KO mice are leaner with increased rates of lipolysis driving insulin resistance, essentially phenocopying ALK7 neutralization[25,62]. It is therefore surprising that the phenotype of acute GDF3 KO is so different from acute ALK7 neutralization or Activin E KO. We find that acute, inducible GDF3 deletion decreases adipose tissue lipolysis and improves insulin sensitivity without changes to body weight—opposite effects to Activin E KO or siALK7. Intriguingly, we find that GDF3 more potently stimulates in vitro signaling through ALK5 than ALK7. It is noteworthy that mice with constitutive adipose-tissue specific deletion of ALK5 (with aP2-Cre) demonstrate markedly improved glucose tolerance[35]. Further investigation into adipose-tissue ALK5 signaling and lipolysis would help to clarify GDF3 signaling.

The actions of TGFβ superfamily proteins affect glucose metabolism and energy homeostasis in diet-induced obesity[64–67]. Here we describe a generalizable effect of TGFβ superfamily proteins on adipocyte lipolysis. We find that BMP signaling and downstream activation of SMAD1/5/8 decrease lipolysis. Conversely, lipid release is increased downstream of SMAD2/3 signaling by TGFβ1. GDF3 is a protein that promotes lipolysis and can both antagonize BMP receptor signaling and activate activin receptor signaling. When we block type 1 receptors (ALK4/5/7) in the SMAD2/3 signaling pathway, GDF3 can no longer affect lipolysis. Yet GDF3 can still increase lipolysis in the presence of an inhibitor of SMAD1/5/8 signaling.

These data suggest that activating SMAD2/3 signaling is likely to affect lipolysis.

Our results highlight the important role that activin and TGFβ proteins play in regulating lipolysis. Specifically, SMAD2/3 activation controls C/EBPα, a transcription factor controlling expression of β3-adrenergic receptor (ADRB3)[68,69]. Adrb3 is selectively expressed in mature adipocytes and responds to catecholamines to promote release of fatty acids and stimulate non-shivering thermogenesis in adipose tissue[70,71]. In this study, we demonstrate that Gdf3-loss of function leads to reduced Adrb3 expression in adipocytes corresponding to reduced rates of lipolysis. Exogenous GDF3 leads to increased adipocyte Adrb3 levels, increased signaling and lipolysis (Fig. 8). Our observations on the effects of recombinant Gdf3 to increase β3- adrenergic signaling are the opposite from the effects of reduced Adrb3 expression seen with Activin B, Activin C, Activin E, and no expression change with Activin A[23,25,62]. Furthermore, a similar pattern is seen with increased Adrb3 levels consistent with elevated lipolysis in global ALK7 KO mice and Ap2Cre driven adipose tissue specific ALK7 knockouts[23]. However, in all these cases, there is good agreement between Adrb3 mRNA levels and rates of adipose tissue lipolysis. Our findings suggest that GDF3 upregulates lipolysis by controlling Adrb3 expression in mature adipocytes. We tested this hypothesis by activating cells with a cAMP analog (8-Br-cAMP) to bypass any effects caused by differential expression of the β3- adrenergic receptor. In response to 8-Br-cAMP, cells lacking GDF3 have identical rates of lipolysis as control cells. These findings suggest that regulation of adipose tissue Adrb3 expression may be the key to adipose tissue control of glucose homeostasis by activin-type ligands.

GDF3 levels are low in lean healthy adult mice but rapidly increase with obesity or ischemia[10,11,13,26,27]. However, acute GDF3 gain-of-function in lean mice led to mild glucose intolerance and insulin resistance with no change in body weight (10). We find that acute Gdf3 knockout in obese mice leads to suppressed adipose lipolysis and lower blood glucose levels leading to anti-diabetic effects. One critical limitation in the field is the absence of a validated method for quantifying circulating levels of GDF3. This is due to the highly conserved primary sequence of ligands, where generating an appropriate anti-GDF3 antibody lacking cross-reactivity to other proteins has proven challenging. Although GDF3 ELISAs are commercially available, our efforts to show specificity in samples from GDF3 KO mice have not been successful.

Our results diverge from previously published work in several critical ways. One previous study demonstrated that Gdf3 decreased rates of lipolysis in primary adipocytes from TSOD mice[13]. We find that Gdf3 increases rates of lipolysis in adipose tissue explants, primary adipocytes, and immortalized adipocytes derived from C57Bl6/J mice. Prior studies had also suggested that GDF3 expressed in adipose tissue macrophages was important for the decreased lipolysis seen in aging mice on a standard diet due to increased catecholamine degradation[14]. However, lipolysis is regulated differently in young and old mice, as well as in lean vs diet-induced obese mice.

In conclusion, our study shows that deletion of the obesity associated gene Gdf3, leads to reduced lipolysis and improved metabolic health in obese mice, with no change in body weight. We also show that acute treatment with recombinant GDF3 can lead to increased stimulated lipolysis in mature adipocytes and these effects of GDF3 are mediated by changes in Adrb3 expression levels in adipocytes. This study, using the first conditional deletion of GDF3, provides new insights to understand the role of GDF3 in the pathogenesis of obesity and metabolic dysregulation. This study further suggests that decoy ACVR2A/2B receptor drugs being considered for anti-obesity co-therapies with GLP1-agonist drugs might additionally involve the suppression of GDF3 signaling and warrant further exploration.

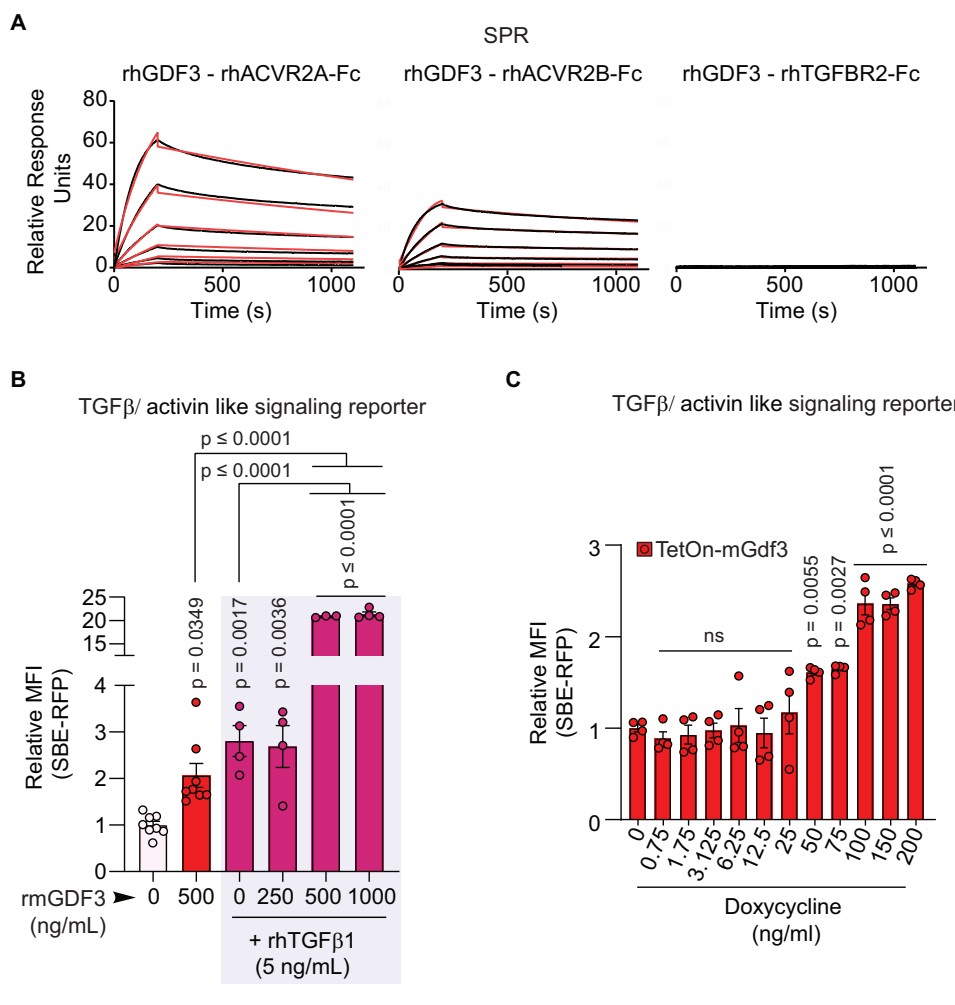

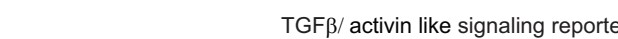

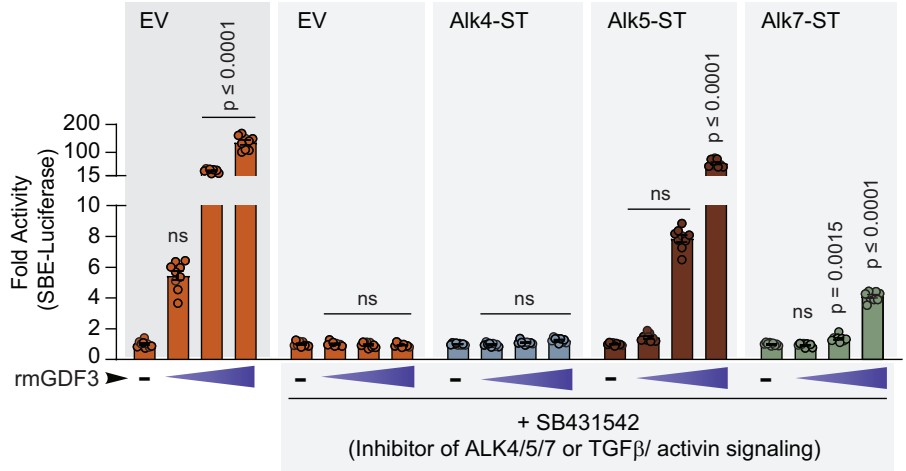

## Methods

### Animal models

All animal experiments were performed with approval from the Institutional Animal Care and Use Committees (IACUC) of The Harvard Center for Comparative Medicine, Beth Israel Deaconess Medical Center. Mice were maintained at 12 h/12 h light/ dark cycles, 22 ± 2 °C room temperature, and 30%–70% humidity with ad libitum access to food and water in individually ventilated cages. Cages and bedding were changed every 2 weeks, and mice were monitored regularly for their health status by animal technicians with the support of veterinary care. They remained free of any adventitious infections for the entire study duration. All breeding mice were fed a special diet ad libitum (25% kcal protein, 13% kcal fat, 62% kcal carbohydrates; Lab Diet, # 5053). After weaning mice were maintained on standard chow diet

**Fig. 6 | GDF3 acts as a TGFβ/activin-like SMAD2/3 signaling agonist. A** SPR analysis of rhGDF3 with rhACVR2A-Fc, rhACVR2B-Fc and rhTGFBR2-Fc. **B** Relative MFI of SBE-RFP in dual reporter preadipocytes incubated in serum free media alone (control) or with rmGDF3 and/or rhTGFβ1 in serum free conditions (Biological replicates: $n = 8$ for control and 500 ng/mL rmGDF3, $n = 4$ for rhTGFβ1 with rmGDF3 0, 250 and 1000 ng/mL, $n = 3$ for rhTGFβ1 and rmGDF3 500 ng/mL). Each n represents the average relative MFI of approximately 1000 sorted live cells).
**C** Relative MFI of SBE-RFP in C2C12 dual reporter myoblasts transiently transfected with the TetOn-Gdf3 vector, followed by 48 h of indicated doses of doxycycline, sorted for YFP, RFP, and BFP triple positive cells ($n = 4$ biological replicates where each n represents the average relative MFI of all triple positive live cells per well).

**D** Relative fold change in SBE-luciferase activity in HEK293T cells expressing either a control empty vector (EV) or vectors encoding type1 TGFβ receptor mutants (Alk4-ST, Alk5-ST or Alk7-ST) that are resistant to SB-431542, the inhibitor of type1 TGFβ/activin-like signaling receptors (ALK4/5/7). All cells were treated with increasing doses of rmGDF3 (0, 250, 500 or 1000 ng/mL) in the absence or presence of SB-431542 ($n = 9$ biological replicates, where each 9 represents the average luciferase activity per well per group). **B–D** Data are presented as mean values +/- SEM, ns = not significant. Statistical comparisons were made using one way ANOVA with Tukey's multiple comparisons test (**B**) or Dunnet's multiple comparisons test (**C, D**). Source data are provided as a Source Data file.

(27% kcal protein, 17% kcal fat, 57% kcal carbohydrates; LabDiet, # 5008) or switched to high-fat diet (HFD, 20% kcal protein, 60% kcal fat, 20% kcal carbohydrates; Research Diets,# D12492i) beginning at ~8 weeks of age for the indicated durations. Mice were always fed ad libitum unless specified otherwise. At the end of experiments, mice were euthanized using $CO_2$, blood samples were collected via cardiac puncture in EDTA coated tubes on ice, tissues were harvested and snap-frozen in liquid nitrogen and stored at −80 °C, until processing, unless otherwise specified.

### Gdf3 floxed mice (Gdf3fl/fl)
Homozygous Gdf3 floxed mice (Gdf3fl/fl) on a C57Bl/6N genetic background were generated using "Easi-CRISPR"[72] with the help of the BIDMC transgenic mouse core. Briefly fertilized C5Bl/6N zygotes were injected with a mixture of sgRNA, a synthetic single stranded DNA (ssDNA) sequence containing the loxp sites flanking the region of interest and Cas9 protein (PNA Bio, # CP01-50). Out of 6 possible F0 founderpups, three were verified by sequencing to have the floxed Gdf3 sequence inserted into the right location and were bred to expand the colony. The sgRNA sequences used flanked exon 1 of the mouse Gdf3 gene (sgRNA 1: ACACTGCTGACCAACCCGCG targeting 73 bp upstream of Exon 1 and sgRNA 2: CTGGGCTGATCTTGGAACCA targeting 121 bp downstream of Exon 1). Exon 1 encoding the Gdf3 start codon (ATG) was targeted such that the entire first exon of Gdf3 would be deleted in the presence of Cre recombinase. Gdf3fl/fl mice were backcrossed to the C57Bl/6 J background for 6 generations before performing any experiments on them.

Whole-body Gdf3 knockout (Gdf3KO) mice were generated by crossing homozygous Gdf3fl/fl mice with hemizygous Rosa-CreERT2/- mice (B6.129-Gt(ROSA)26Sortm1(cre/ERT2)Tyj/J, The Jackson Laboratory, Stock # 008463) followed by tamoxifen injections[28]. Homozygous Gdf3fl/fl mice without the Rosa-CreERT2 transgene were used as controls for all experiments. 8 week old adult male or female Gdf3fl/fl and Gdf3fl/fl::Rosa-CreERT2/- mice were fed HFD ad libitum for 8 weeks. Intraperitoneal (IP) injections of tamoxifen in 100% Corn oil (75 mg/kg body weight, # T5648; Millipore Sigma, St. Louis, MO) were administered daily for seven consecutive days to activate Cre recombinase and achieve whole body deletion of Gdf3. The mice were allowed to recover from tamoxifen injections for 4 weeks. Tissue samples were harvested at the indicated time points and snap-frozen in liquid nitrogen before storage at −80 °C for further processing.

AAV mediated Gdf3AKO mice were generated by feeding adult male Gdf3fl/fl mice HFD for 8 weeks and then bilaterally injecting them with either AAV2/DJ-CMVeGFP (AAV-GFP) $1.13 \times 10^{13}$ vg/ml or AAV2/DJ-CMVCre-wtIRESeGFP (AAV-Cre) $1.3 \times 10^{13}$ vg/ml at a dose of 5ul each into the eWAT. Mice were provided long acting Buprenorphine SR LAB as an analgesic and isofluorane-induced anesthesia for all surgical interventions. The AAVs were obtained from the University of Iowa Viral Vector Facility, AAV-GFP (VVC-U of Iowa-4382) and AAV-Cre (VVC-U of Iowa-5714). The mice were allowed to recover from surgery for

2 weeks. Tissue samples were harvested 4 weeks after AAV injections and snap-frozen in liquid nitrogen before storage at −80 °C for further processing.

### Mouse genotyping
Mouse ear clips or tails were lysed in 100 μL tail lysis buffer (0.025 N sodium hydroxide, 2 mM EDTA) at 99.9 °C for 30 mins and then neutralized with 100 μL of neutralization buffer (0.04 M Tris-HCl, pH 7.5). Genotyping reactions were performed with OneTaq Master mix (New England Biolabs Inc, # M0482L) and 5 μM each of primers Rosa F3 (AAAGTCGCTCTGAGTTGTTAT), Rosa F4 (GGAGCGGGAGAAATGGA-TATG) and Rosa R3 (CCTGATCCTGGCAATTTCG) for the Rosa-CreERT2 gene and primers Gdf3flox-F (AGATGGACACCATGGCTGGC) and Gdf3flox-R (GTCTAGCTCTCGGAATGCTG) for the floxed Gdf3 gene. Expected band sizes were 650 bp and 800 bp for the wildtype and Cre band for the RosaCreERT2 reaction and 200 bp and 235 bp for the wildtype and Gdf3 'flox' bands for the Gdf3flox reactions.

### Continuous glucose monitoring (CGM)
CGM was performed with mouse-sized HD-XG probes from Data Sciences International (DSI) as described in ref. 31 and following the manufacturer's recommendation. Adult Gdf3fl/fl male mice were fed HFD for 8 weeks and the glucose sensor was surgically implanted in their aortic arch while simultaneously injecting AAV-GFP or AAV-Cre into their eWAT bilaterally. Of the 10 HFD Gdf3fl/fl mice undergoing surgery, 6 survived the CGM implantation procedure ($n = 3$ for each group). Mice were allowed to recover for 2 weeks post-surgery before any experimental interventions. Following surgical recovery, mice were placed into a Promethion indirect calorimeter and glucose probes were powered on by proximity to a magnetic field. After 4 h of acclimation to the Promethion, probes were calibrated with a 2-point calibration at ad libitum baseline and 20 min following a 2 g/kg body weight glucose bolus. The tail tip was nicked to elicit a drop of blood for calibrations, and the average of two measurements with a Contour Next EZ glucometer (Bayer) was used. Single-point calibrations were taken twice weekly from ad libitum fed mice for the experiment. The calibration measurements were repeated if the glucometers showed more than 10% difference in glycemia values between each glucometer. For the purpose of comparison, data from chow-fed mice was analyzed from Rubio et al.[31]. Glucose data were analyzed in R version 4.3.0 and the Iglu package version 3.4.2[73,74].

### Glucose and insulin tolerance tests
For the glucose tolerance test (GTT), mice were fasted for 4 h and injected IP with a glucose bolus (1 g/kg body weight). Glycemia was measured at 0, 15, 30, 60, and 120 min post glucose injections using a Contour Next EZ glucometer (Bayer) or continuously every 1 min via the CGM probes. For the insulin tolerance test (ITT), mice were fasted for 4 h during the light phase, followed by IP injections of recombinant human insulin (Humulin R, Eli Lilly, # NDC 0002-8215-01) in saline at

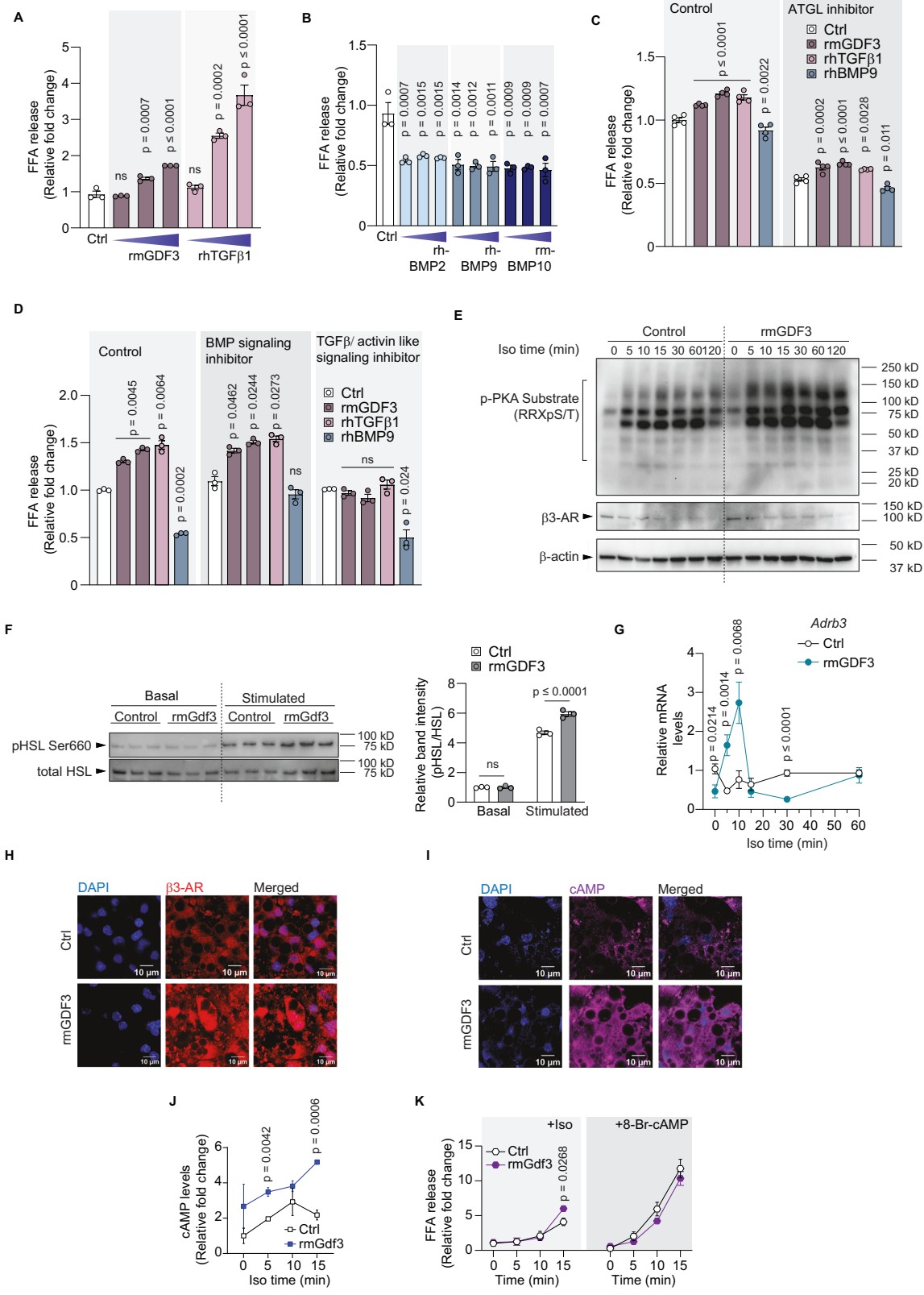

0.75 U/kg body weight. Blood glucose levels were measured at 0, 15, 30, 60, and 120 min after insulin injection.

**Plasma insulin**
Blood samples were collected from lateral tail vein bleeds after a 4 h fast in EDTA coated tubes, centrifuged for 15 min, at 2000g, at 4 °C, and the resulting supernatant was collected as plasma and stored at

−80 °C until further processing. Plasma insulin concentrations were measured with the Ultra-Sensitive Mouse Insulin ELISA Kit (Crystal Chem, # 90080) as per manufacturer's instructions.

**Indirect calorimetry/metabolic phenotyping**
For measurements of metabolic rate, data was collected from individually caged mice placed in the Promethion indirect calorimeter

**Fig. 7 | Recombinant GDF3 enhances lipolysis in vitro in cultured mature adipocytes. A–K** Immortalized iWAT-SVF cells differentiated into mature adipocytes. **A, B** FFA release following 15 min of isoproterenol stimulation in cells pretreated with recombinants for 24 h. **A** Recombinants: rmGDF3 (250, 500, 1000 ng/mL); rhTGFβ1 (2.5, 5, 10 ng/mL), (*n* = 3). **B** Recombinants: rhBMP2, rhBMP9, rmBMP10 (63, 125, 250 ng/mL each), (*n* = 3). **C, D** Relative FFA release from adipocytes pretreated without (control) or with a specific inhibitor for 1 h, incubated with recombinants for 24 h before isoproterenol stimulation for 2 h. **C** Inhibitor: Atglistatin (100 μM). Recombinants: rmGDF3 (500, 1000 ng/mL left to right); rhTGFβ1 (2.5 ng/mL); rhBMP9 (125 ng/mL); (*n* = 4). **D** Inhibitors: LDN193189 (BMP signaling inhibitor, 0.5 μM) or SB-431542 (TGFβ/activin-like signaling inhibitor, 5 μM) Recombinants: rmGDF3 (500, 1000 ng/mL); rhTGFβ1 (2.5 ng/mL); rhBMP9 (125 ng/mL); (*n* = 3). **E–K** Adipocytes pretreated with serum free media (Control) or rmGDF3 (500 ng/mL) for 24 h. **E, F** Western blot analysis (**E**) Protein levels of p-PKA substrates, β3-AR, and β-actin at indicated time points post isoproterenol

stimulation (Iso time, *n* = 1 per group per time point). **F** Left: Protein levels of pHSL Ser 660 and total HSL. Right: Relative band intensities of pHSL/total HSL in basal and 15 min isoproterenol stimulated conditions (*n* = 3 biological replicates per group). **G** RTqPCR analysis of *Adrb3* following isoproterenol stimulation (*n* = 6). **H** Immunofluorescence (IF) staining of β3-AR (red) after 5 min of isoproterenol stimulation. Nuclear stain: DAPI (blue). **I** IF staining of cyclic AMP (purple) 15 min post isoproterenol treatment. Nuclear stain: DAPI (blue). **J** Relative cAMP levels in adipocytes following isoproterenol stimulation for indicated time points. **K** FFA release in response to isoproterenol (left) or the cAMP analog, 8-Bromo-cAMP (right) treatment. **A–G, J, K** Data are presented as mean values +/- SEM, ns = not significant. Statistical comparisons were made using one way ANOVA with Dunnett's multiple comparisons test (**A–D**) or two way ANOVA with Sidak's multiple comparisons test (**F**) or unpaired two-tailed Student's *t*-tests for each time point (**G, J, K**). Source data are provided as a Source Data file.

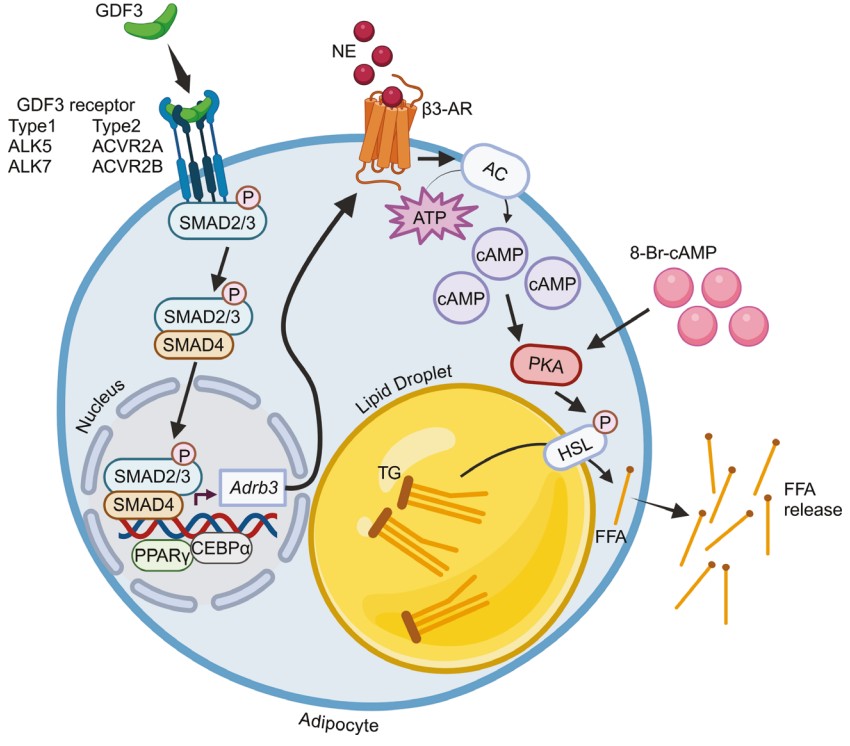

GDF3 - Growth differentiation factor 3; β3-AR - Beta-3 Adrenergic Receptor; AC - Adenylyl Cyclase;
FFA - Free fatty acid; TG - Triglyceride; HSL - Hormone Sensitive Lipase; NE - Norepinephrine

**Fig. 8 | Graphical representation of GDF3's mechanism of action on adipocytes.** GDF3 binds to its heteromeric receptors on the adipocyte cell surface, which triggers a cascade of signaling responses leading to the phosphorylation of Smad2 and 3. Phospho-Smad2/3 then bind to Smad4, enter the nucleus and bind to the promoter region of the gene Adrb3 which encodes the β-3-adrenergic receptor (β3-AR), priming the cell to increase Adrb3 expression in response to thermogenic stimulation through norepinephrine (NE) or its analogs like Isoproterenol. Upon thermogenic stimulation, adipocytes pre-exposed to GDF3 increase β3-AR

expression which leads to increased cellular cAMP levels because of increased adenylyl cyclase (AC) enzyme activity. Increased cAMP levels lead to increased protein kinase A (PKA) activity which leads to the phosphorylation of downstream targets like hormone sensitive lipase (HSL), located on the surface of adipocyte lipid droplets. This leads to the breakdown of adipocyte triglycerides (TG) into free fatty acids (FFA) that are released from the cells. Created in BioRender. Banks, A. (2025) https://BioRender.com/v394jid.

system (Sable Systems International) with temperature-controlled cabinets. Mice were provided with ad libitum access to food and water, unless specified otherwise. If mice were implanted with CGM probes, DSI telemetry receiver bases were placed beneath the Promethion Cages to match the probes. Mice were maintained under 12 h/12 h (0600/1800) light/dark cycles at an ambient temperature of 23 ± 0.2 °C unless otherwise specified. Position and physical activity were collected every second. Rates of oxygen consumption (VO₂) and

carbon dioxide production (VCO₂) were measured every 2 min. Data was imputed to a 1 min resolution to match the glucose and body temperature recordings. Energy expenditure was calculated with the Weir equation[75].

**In vivo lipolysis assays**
Mice were fasted for 4 h and injected IP with isoproterenol (10 mg/kg body weight). Plasma samples from lateral tail vein bleeds were

collected before and 15 min after isoproterenol injections, as described above. Non-esterified free fatty acid (NEFA) plasma concentrations were quantified using the Free Fatty Acid Quantitation Kit (Sigma-Aldrich # MAK044, or Abcam # ab65341).

## Ex vivo lipolysis assay

Mice were euthanized as described above and bilateral iWAT and eWAT depots were collected and weighed. Approximately 50 mg each of iWAT and eWAT were transferred into separate wells of 24 well plates containing 1 mL of pre-warmed Krebs-Ringer Bicarbonate HEPES (KRBH) buffer, pH 7.4, per well. Once all the samples were collected, using a pair of sterile scissors, adipose tissue samples were cut into smaller fragment and the media in each well was replaced with 1 mL of pre-warmed KRBH buffer containing 2% fatty acid-free bovine serum albumin (KRBH-BSA) per well. Media from each well was aspirated and replaced with 1 ml of prewarmed KRBH-BSA buffer containing iso-proterenol (10 µM). The plates were incubate at 37 °C for 5 mins. 100 µL of media was collected from each well as the baseline. 100 µL of fresh KRBH-BSA buffer containing 10 µM isoproterenol was added to each well. The plates were incubated 37 °C for 2 h following which all the media was collected from each well. All the baseline and 2 h media samples were incubated at 65 °C for 10 min to inactivate any residual enzymatic activity in the media. These samples were then stored at −80 °C until further processing. Tissue samples were washed with PBS and then lysed in RIPA buffer for protein quantifications using the Pierce BCA protein assay kit (Thermo Fisher, # 23225) following the manufacturer's instructions. FFA media concentrations were analyzed using the Free Fatty Acid Quantitation Kit (Sigma-Aldrich # MAK044, or Abcam # ab65341) following the manufacturer's instructions and normalized to protein levels[76].

## RNA Isolation and RT-qPCR

RNA was extracted using the Direct-zol RNA miniprep kit (Zymo Research, # R2050) following the manufacturer's instructions. Real-time quantitative PCR (RT-qPCR) was performed using cDNA generated with the High Capacity cDNA Reverse Transcription kit (ThermoFisher Scientific, # 4368813) with SYBR Select Master Mix (Applied Biosystems, # 4472920) on the QuantStudio 6 Real-Time PCR System (Applied Biosystems). Relative mRNA expressions were determined by the 2^-ddCt method normalized to TATA-binding protein (TBP) levels. All primer sequences used are listed in Table S3.

## SVF isolation from iWAT

The stromal vascular fraction (SVF) containing preadipocytes and mononuclear cells was extracted from inguinal white adipose tissue (iWAT) of 8–12 week old female Gdf3^fl/fl mice as described previously (Hui Yu et al. 2018). Briefly, mice were euthanized using $CO_2$ as described above and the iWAT was isolated, minced and digested in PBS containing 1.5 U/mL collagenase D (Sigma-Aldrich, # 11088882001), 2.4 U/mL dispase II (Sigma-Aldrich, # 4942078001), and 10 mM CaCl2 for 45 min in a 37 °C shaking water bath. The digestion enzymes were neutralized with SVF culture medium, DMEM/F12 (Gibco, # 10565042) supplemented with 10% heat-inactivated fetal bovine serum (iFBS, Gibco, # 10082147), and 1X Penicillin-Streptomycin (pen-strep, Gibco; # 15070063). Digested tissue was filtered sequentially through 100 mm and 40 mm cell strainers, centrifuged and the resultant pellet was resuspended in SVF culture media and plated onto 100 mm cell culture treated plates and grown in cell culture incubators at 37 °C, with 5% $CO_2$. These cells were passaged every few days and cells between the 2nd and 4th passage cycle were used to generate differentiated primary adipocytes.

## Cell lines

HEK293T cells (human female in origin) were cultured every 3–4 days using 0.05% Trypsin EDTA (Gibco, # 25300054) and maintained in DMEM (Gibco, # 11965118) supplemented with 10% iFBS and 1X pen-strep. C2C12 myoblasts (ECACC, # 91031101, C3H murine female in origin) were cultured every 2- 3 days using 0.25% Trypsin-EDTA (Gibco, # 25200114) and maintained in DMEM (GIBCO, 11965118) supplemented with 20% iFBS and 1X pen-strep. Immortalized iWAT-SVF cells were a gift from Shingo Kajimura's lab[39]. These cells were cultured every 3–4 days using 0.25% Trypsin-EDTA, and maintained in DMEM containing 10% iFBS and 1X pen-strep. All cell lines were maintained under sterile conditions at 37 °C and 5% $CO_2$. HEK-293 cells (RRID: CVCL_0045, human female in origin) stably expressing the gene for firefly luciferase downstream of the $CAGA_{12}$ promoter (SBE-Luciferase)[77] were cultured every 3–4 days using 0.05% Trypsin EDTA and maintained in DMEM with 10% iFBS, 1X pen-strep and 0.5 mg/mL Puromycin.

## In vitro adipocyte differentiation

Immortalized or primary preadipocytes were plated and grown to confluency in regular cell culture treated or collagen coated plates respectively. Differentiation was initiated (day1) with the appropriate complete media containing the induction cocktail of 0.5 mM 3-isobutyl-1-methylxanthine (IBMX, Sigma-Aldrich, # I5879), 2 µg/ml dexamethasone (Sigma-Aldrich, # D4902-100MG), 125 µM indomethacin (Sigma-Aldrich, # I7378), and 0.5 µM rosiglitazone (Sigma-Aldrich, # R2408) for 2 days. The cells were then switched to maintenance media containing 5 µg/ml insulin (Sigma-Aldrich, # I6634) and 0.5 µM rosiglitazone, which was refreshed every other day until day 7 or until 90 to 100% of cells were differentiated. Differentiated primary adipocytes from Gdf3^fl/fl and Gdf3^fl/fl::RosaCre^ERT2/− mice were incubated with 25 µM 4-hydroxy tamoxifen (4OHT, Sigma-Aldrich, # H7904) in maintenance media for 72 h to induce Cre recombinase activity to generate control Gdf3^fl/fl and Gdf3^KO cells. After 72 h cells were switched back to maintenance medium for another 72 h to wash out the 4OHT prior to any experimental intervention.

## In vitro lipolysis assays

Primary differentiated Gdf3^fl/fl and Gdf3^KO adipocytes were generated in 48-well collagen coated plates as described above and serum starved in DMEM/F12 with 1X pen-strep and 0.3% fatty acid free BSA for 4 h.

Immortalized preadipocytes, were differentiated into mature adipocytes, in 48-well plates as described above, serum starved for 4 h in DMEM containing 1X pen-step and 0.3% fatty acid free BSA, followed by incubations with recombinant proteins in serum-free media for 24 h prior to initiating lipolysis. For lipolysis assays including different inhibitors, cells were serum starved for 3 h, followed by incubations with 100 µM Atglistatin (Selleckchem, # S7364) or 0.5 µM LDN-193189 (Selleckchem, # S2618) or 5 µM SB-431542 (Selleckchem, # S1067) for 1 h in serum free media, followed by incubations with recombinant proteins in serum-free media for 24 h prior to initiating lipolysis. The details of recombinant proteins used are listed in Table S1.

Media was then aspirated from all wells and replaced with KRBH-BSA buffer. Baseline samples were collected immediately from all the wells and replaced with KRBH-BSA buffer containing isoproterenol (10 µM). Cells were incubated for the indicated time points in a 37 °C incubator following which media was collected (stimulated samples). All samples were stored at −80 °C until further processing.

For measuring FFA levels all the samples were thawed at room temperature and incubated at 65 °C for 10 min to inactivate any residual enzymatic activity. FFA levels were quantified using the Free Fatty Acid Quantitation Kit (Sigma-Aldrich # MAK044, or Abcam # ab65341) following the manufacturer's instructions.

## Plasmid vectors

The pHK3-BRE-Citrine (BRE-YFP) fluorescent reporter, as well as the piggyback transposase (p-base plasmid), were gifts from Michael B.

Elowitz's lab (Antebi et al. Cell, 2017). To generate the pHK3-CAGA12-mCherry (SBE-RFP), the BRE-Citrine was replaced with CAGA12-mCherry using the NEBuilder® HiFi DNA Assembly Cloning Kit (New England Biolabs, # E5520S). For ease of readability, these reporters are referred to as BRE-YFP and SBE-RFP in the manuscript.

The pTetON-TRE-TagBFP2 and pTetON-TRE-mGdf3-P2A-TagBFP2 plasmids (referred to as TetOn-Control and TetOn-mGdf3 in the manuscript) were synthesized with codon optimization of mouse Gdf3 cDNA by Vector Builder Inc. GDF3 and/or TagBFP2 expressions were induced using the indicated concentrations of doxycycline hyclate (Sigma-Aldrich, # D5207) for the indicated times. All transfections were carried out using Lipofectamine 3000 (Invitrogen, # L3000001) following manufacturer instructions.

### Western blot analysis
HEK293T cells were plated in collagen-coated 6-well plates to reach 70–90% confluency overnight. These cells were then transfected with 2 μg/ well of either the TetOn-Control or TetOn-mGdf3 vectors using Lipofectamine 3000 (Invitrogen, # L3000015) following the manufacturer's protocol. 24 h post-transfection, the cells were incubated in serum-free media with or without doxycycline for 48 h before lysis for protein extraction.

Immortalized or primary differentiated adipocytes were generated as described above, in 6-well regular or collagen coated plates respectively. Mature adipocytes from immortalized preadipocytes were serum starved for 4 h followed by incubations with recombinant proteins in serum-free media for 24 h. Mature primary Gdf3$^{fl/fl}$ and Gdf3$^{KO}$ adipocytes were serum starved for 4 h, prior to stimulating lipolysis. Cells were then incubated in KRBH-BSA buffer with our without isoproterenol (10 μM) for the indicated time points in a 37 °C incubator.

Whole-cell extracts were prepared by washing the cells twice with PBS and then scraping the cells in RIPA lysis buffer (50 mM Tris, pH 7.5, 150 mM NaCl, 1% NP-40, 0.5% sodium deoxycholate, 0.1% SDS) containing 1X HALT protease and phosphatase inhibitor cocktail (Thermo Fisher, # 78440). Protein concentrations were quantified using the Pierce BCA protein assay kit (Thermo Fisher, # 23225) as per manufacturer's instructions, and protein samples were prepared in reducing Laemmli buffer and heated for 10 min at 95 °C. Equal amounts of protein were loaded onto precast Mini-PROTEAN TGX gels (Bio-Rad), separated by SDS-PAGE using the appropriate Bio-Rad apparatus and transferred to PVDF membranes using the Trans-Blot Turbo transfer system (Bio-Rad). The blots were blocked with 5% BSA in TBST and blotted according to the manufacturer's recommendations for the indicated primary and secondary antibodies (Table S2), followed by development with the femtoLUCENT™ PLUS-HRP Chemiluminescent kit (G-Biosciences, # 786-003) or the SuperSignal™ West Pico Chemiluminescent kit (Thermo Fisher, # 34075) and the BioRad Chemidoc Touch Imaging system. The details of recombinant proteins and antibodies used are listed in Tables S1 and S2. Uncropped and unprocessed scans of all western blots have been provided in the Source Data File.

### Immunofluorescence staining and confocal microscopy
Briefly, differentiated cultured and primary adipocytes were serum starved for 4 h. Cultured adipocytes were incubated with recombinant proteins in serum-free media for 24 h. Cells were then incubated in KRBH-BSA media with or without isoproterenol (10 μM) for the indicated time points, washed with PBS, fixed for 15 min with 4% PFA in PBS, followed by permeabilization for 10 min with 0.1% Triton X-100 in PBS. Cells were then washed with PBS, blocked with 4% BSA in PBS for 1 h at room temperature, and incubated with different primary antibodies diluted in the blocking solution for 12–16 h at 4 °C temperature. Adipocytes were washed with PBS (3 × 10 min), incubated for 1 h at room temperature with secondary antibodies, and mounted in Pro-Long Gold antifade reagent (Molecular Probes, Milipore Sigma, #

P36965) with DAPI (Sigma Aldrich, # D9542). Three independent images in separate locations were collected for each sample ($n = 3$ biological replicates per group). Representative images were presented in the manuscript. Recombinant Proteins and primary and secondary antibodies used are listed in Tables S1 and S2. Immunostaining data were collected using the Zeiss Confocal System and analyzed using Fiji software 1.54 f.

### cAMP levels
Briefly, either immortalized preadipocytes were differentiated or primary preadipocytes were differentiated into mature adipocytes in regular or collagen coated 48-well plates respectively, as described above. Cells were serum starved for 4 hr. Immortalized cultured adipocytes were additionally incubated with recombinant proteins (Table S1) in serum-free media for 24 h. Media was then aspirated and replaced with KRBH-BSA buffer with and without containing isoproterenol (10 μM) for respective time points (0-15 min). After the washout of 4OHT, primary adipocytes were serum starved for 4 h, and lipolysis was induced with isoproterenol (10 μM) with respective time points. After lipolysis stimulation with isoproterenol adipocytes were lysed of 0.1 M HCl. Determination of cAMP was performed using cAMP ELISA Kit (Cayman chemicals, # 581001) following the manufacturer's instructions.

### Generation of dual reporter cells
Immortalized iWAT-SVF preadipocytes or C2C12 myoblasts were cotransfected with the BRE-YFP reporter plasmid and the piggyback transposase (p-base plasmid) designed to be stably integrated using the piggybac transposon system[78]. Stable cells were generated by selection with 500 μg/mL or 800 μg/mL of Hygromycin B (Sigma-Aldrich, # H7772) respectively. The cells were then co-transfected with the SBE-RFP reporter and the p-base plasmid. Cells were selected for stable integration of both reporters by two rounds of fluorescence-assisted cell sorting for YFP and RFP double-positive cells (dual reporter cells).

### Flow cytometry
Immortalized preadipocyte dual reporter cells were plated in a 96-well cell culture plate to reach 50–70% confluency the next day, in complete media (DMEM high glucose with 10% iFBS and pen-strep. Cells were then incubated with the specified concentrations of recombinant proteins (Table S1) in complete media for 24 h. If cells needed to be serum starved, the media was replaced with serum starvation medium for 4 h the day after plating, followed by incubation with recombinant proteins in serum-free media for 24 h.

For genetically encoded Gdf3 and/or BFP induction, C2C12 dual reporter cells were transiently transfected with either the TetOn-Control or the TetOn-mGdf3 plasmids in 10 cm cell culture dishes. After overnight incubation in complete medium, these cells were split into a 96-well cell culture plate and cultured overnight in complete media. Cells were serum-starved for 4 h and then treated with the indicated doses of doxycycline hyclate (Sigma-Aldrich, # D5207) in serum-free media for 48 h at the indicated concentrations. The cells were then processed for flow cytometry.

Briefly, the 96 well plate media was removed, and cells were washed with PBS. Cells were then dissociated with 0.25% Trypsin EDTA in a cell culture incubator for approximately 5 min. Trypsin was neutralized with PBS containing 3% iFBS, and the plate was spun in a refrigerated centrifuge at 500 g for 10 min with a swinging rotor. The supernatant was removed, and FACS buffer (PBS + 0.3% BSA) containing 1:10000 LIVE/DEAD Fixable Far Red Dead Cell Stain (Thermo Fisher Scientific, # L34973) was added to the cells for 15 min on ice. The plate was then centrifuged again; the media was removed, washed with FACS buffer, and centrifuged again. The supernatant was removed, and each well was incubated with 200uL of FACS buffer. These cells

were then analyzed using the CytoFLEX flow Cytometer (Beckman Coulter) and the CytExpert acquisition and analysis software. Briefly, single unstained cells and single fluorescent controls were used to set the gains and gates for each fluorescent channel. Cells were gated to analyze 1000 live single cells from each well. The mean fluorescence intensity (MFI) of either YFP or RFP of the live cells from each well was used for the subsequent analysis. For the doxycycline-induced BFP or GDF3 and BFP expression experiments, the MFI of YFP or RFP was measured from cells expressing YFP, RFP, and BFP.

### Surface Plasmon Resonance (SPR)

SPR was performed in 20 mM HEPES pH 7.4, 350 mM NaCl, 3.4 mM EDTA, 0.05% P-20 surfactant, 0.5 mg/ml BSA at 25 °C on a Biacore T200 optical biosensor system (Cytiva). Human Fc-fusion constructs of each receptor were purchased from R&D Systems (Table S1). Each receptor was captured using a series S Protein A sensor chip (Cytiva) and immobilized with a target capture level of ~70 RU with a contact time of 60 s at 20 μL/min flow rate. A 16-point, two-fold serial dilutions were performed for rhGDF3 (R&D systems, Table S1) at a starting concentration of 12.5 nM–0.391 nM. Each cycle had an association of 200 s and a disassociation of 900 s at a flow rate of 50 μL/min. The sensor chip was regenerated with 10 mM Glycine pH 1.7. Kinetic analysis was utilized on the Biacore T200 evaluation software and fit by 1:1 fit model with mass transport limitations to determine the association rate ($k_a$) and dissociation rate ($k_d$) for each ligand:receptor pair; $K_D = k_d/k_a$. All SPR experiments were performed in triplicate, except TGFβRII-Fc, which was run in duplicate and where no binding was observed. Curves were fit individually using a 1:1 model for kinetic binding using Biacore software and plotted with GraphPad Prism. The recombinant proteins used are listed in Table S1.

### Biolayer Interferometry (BLI)

All binding assays were performed with the Gator Plus (Gator Bio) instrument using 96-tilted well plates. Anti-Human IgG Fc Gen II (# 160024) and Anti-Mouse IgG Fc (# 160004) probes, BLI 96-Flat Plate Polypropylene (# 130150), Max Plate (# 130062), and Regen Buffer no salt (# 120063) were all purchased from Gator Bio. Samples were diluted in freshly prepared and filtered running buffer containing 1x phosphate buffered saline, 1 mg/mL BSA, and 0.1% Tween20, pH 7.4 at 25 °C with an orbital shake speed of 1000 rpm. Kinetic assays were performed by first equilibrating the sensors for 600 s in kinetic buffer, followed by 150 s of baseline, 200 s of loading, 150 s of washing, 300 s of association, 300 s of dissociation, and 5 cycles of regeneration (5 s regeneration/5 s neutralization). All the capturing receptors and antibodies were used at a final 10 μg/mL concentration. The target proteins were used at different concentrations, as described in the results section and figure legends. The data were analyzed using the proprietary software offered by Gator Bio®. The recombinants and antibodies used are listed in Tables S1 and S2.

### Isothermal Titration Calorimetry (ITC)

The incremental ITC binding experiments were performed in the Nano ITC Standard Volume (TA Instruments) with a fixed cell by titrating 250 ng/mL of rhBMPR2-Fc, rhBMP2, rmBMP4 or rhBMP9 respectively into 250 ng/mL of rmGDF3. The incremental ITC experiments consisted of 10, 8 μL injections at 350 s intervals with a stirring speed of 300 revolutions per minute (rpm). The data was acquired using ITC Run data acquisition software. ITC data was analyzed using Nano Analyze Software. The recombinant proteins are listed in Table S1.

### Luciferase reporter assay

HEK-293 cells stably expressing the gene for firefly luciferase behind the CAGA$_{12}$ promoter (CAGA-Luc) were plated in 96-well plates at a concentration of 20,000 cells/well. Cells were cultured for 24 h in growth medium, then transiently transfected with either 50 ng empty vector (EV) or 40 ng EV and 10 ng of a plasmid containing the sequence for Alk4-ST, Alk5-ST, or Alk7-ST using the Mirus LT-1 transfection reagent according to the manufacturer's protocol. These ST constructs contain the full-length receptor sequence with a single point mutation making the receptors resistant to inhibition by the small molecule SB-431542 (pRK5 rat ALK5 S278T, pcDNA3 rat ALK4 S282T, pcDNA4B human ALK7 S270T)[79]. Empty pRK5 vector was used as the control. 24 h post transfection, the growth medium was replaced with serum-free medium containing 0.1% BSA and supplemented with varying concentrations of rmGDF3 with or without 10 μM SB-431542. After an 18 h incubation, cells were lysed with Promega Passive Lysis Buffer (Promega # E1941), and luminescence was measured using the Promega Luciferase Assay System (Promega # 1501) and a BioTek Synergy H1 hybrid plate reader.

### Oil red O staining

Lipid accumulation in adipocytes was detected by Oil Red O (ORO) staining. Briefly, the medium was removed, and cells were washed with PBS and fixed in 4% paraformaldehyde for 20–30 min, followed by washing with PBS three times. After PBS was removed, 500 μl of 50% ethanol was added and washed for 15–20 s to remove water. Following that, 200 μl of freshly prepared ORO solutions was added to the wells and incubated for 10–15 min at room temperature. ORO solution was pipetted out, cells were washed with 500 μl of 50% isopropanol for 15–20 s to remove excess dye solution and then washed with PBS more than 3 times until the liquid was clarified; images were captured under a microscope within 24 h before quantitative analysis[80].

### Statistical analyses

Data analysis and plots were generated using GraphPad Prism software or the R programming language version 4.3.0 using the tidyverse package[74,81]. Indirect calorimetry data were exported with Macro Interpreter, macro 13 (Sable Systems) prior to analysis in CalR version 1.3[82]. RER was calculated as $VCO_2/VO_2$. Telemetry recordings contained short periods of signal dropout. Missing values were imputed using the imputeTS R package[83]. As indirect calorimetry and CGM were recorded on the same PC, data files were aligned by the system time to the nearest minute. Data visualization used a 5–15 min rolling mean function to reduce visual noise. All data are expressed as mean values ± SEM unless specified otherwise. All data were assumed to have a normal distribution, and two-tailed Student's $t$-tests, one-way ANOVA with Newman-Keuls Multiple Comparison tests, or two-way ANOVA were used to compare means between groups; $p < 0.05$ was considered significant.

### Reporting summary

Further information on research design is available in the Nature Portfolio Reporting Summary linked to this article.

## Data availability

All data generated in this study are provided within the paper or in the Supplementary information or the Source data file unless specified otherwise. Large data sets from indirect calorimetry, isothermal calorimetry, surface plasmon resonance and biolayer interferometry do not have a suitable public repository for deposition and will therefore be made available upon request. For access requests, please contact corresponding author Dr. Alexander S. Banks (asbanks@bidmc.harvard.edu). Please assume a timeframe of 1 month to receive a response to requests. No restrictions will be imposed on data use via data use agreements. Source data are provided with this paper.

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

## Acknowledgements

Financial support for this project was provided to ASB by the NIDDK (R01DK107717, R01DK133948) and the NIH Office of the Director (S10OD028635). We thank the BIDMC Transgenic Core Facility and Anuradha Rajendran for their help with generating the Gdf3^fl/fl mouse colony. We thank Michael B. Elowitz from Caltech for the pHK3-BRE-Citrine (BRE-YFP) fluorescent reporter, as well as the piggyback transposase (p-base plasmid). We thank Viet Le and Tim Springer from Boston Children's Hospital & Harvard Medical School for helpful discussions and for the gift of recombinant human TGFβ1. We thank Christopher Auger, Anthony Verkerke and Hiroshi Nishida from Shingo Kajimura's lab at BIDMC for help with the Isothermal Calorimetry assays, for the kind gift of the immortalized iWAT preadipocyte cell line and for help with confocal imaging, respectively. We also thank the BIDMC flow core for their help with flow cytometry. Graphical representations in Figs. 1A, 2A, 4C, D and 8 were created in BioRender.

## Author contributions

N.K.: Conceptualization, Methodology, Investigation, Formal Analysis, Writing—Review and Editing, Visualization, Project administration. D.R.: Conceptualization, Methodology, Investigation, Formal Analysis, Writing- Original draft, Writing—Review and Editing, Visualization. D.V.: Investigation. W.B.R.: Software, Investigation, Writing—Review and

Editing. G.R.G.: Conceptualization, Methodology, Investigation, Formal Analysis, Writing—Review and Editing, Visualization. L.T.: Methodology, Investigation, Formal Analysis, Writing—Review and Editing, Visualization. K.V.: Methodology, Investigation, Formal Analysis, Writing—Review and Editing, Visualization. D.E.M.: Conceptualization, Resources. V.R.: Conceptualization, Supervision. P.B.Y.: Conceptualization, Supervision. T.B.T.: Conceptualization, Supervision. A.S.B.: Conceptualization, Methodology, Investigation, Formal Analysis, Writing—Original draft, Writing—Review and Editing, Visualization, Data Curation, Project administration, Supervision, Project Funding Acquisition.

## Competing interests

ASB receives research funding from Eli Lilly and Company. VR is a consultant and on the scientific advisory board for Keros Therapeutics. PBY is a co-founder, consultant, and stockholder for Keros Therapeutics. PBY is a co-founder and stockholder of Modal Therapeutics, and OrphAI Therapeutics. PBY receives research funding from Gossamer Bio, Inc., and Pfizer, Inc. ASB and TBT are consultants for Alnylam Pharmaceuticals and Keros Therapeutics. The authors have no other relevant competing interests to declare.
