## [Peer Review file · Nature Communications]

Acute regulation of murine adipose tissue lipolysis and insulin resistance by the TGF β superfamily protein GDF3

Corresponding Author: Professor Alexander Banks

Version 0:

Reviewer comments:

Reviewer #1

(Remarks to the Author)

The authors have adequately addressed my queries from the first review cycle. I also feel that they have addressed the points raised by the other reviewers.

There are still numerous questions that could be addressed on molecular mechanisms and potential GDF3 effects on other cell types, but the authors contribute considerable new findings to the field, including technological innovations (new GDF3 KO mouse model ; new reporter assay for BMP v TGF signaling) as well as revealing a new GDF3-ACVR2A/B-ALK1 signaling pathway and unexpected activities of GDF3 KO in enhancing metabolic health through PPAR γ stimulated lipolysis.

GDF3 is expressed in embryonic stem cells, but in the adult, it is mainly expressed from adipocytes of obese animals. The authors show that adipocyte-specific GDF3 knock out (KO) when induced in adult obese mice, in contrast to total GDF3 KO present from early embryogenesis, has no effect on animal weight, but enhances metabolic health of the mice in terms of better glucose tolerance and reduced circulating insulin levels on fasting. Surprisingly they saw reduced adipose tissue lipolysis, consistent with no reduction in fat mass.

Although much has been published on GDF3 signaling and regulation of obesity, as eloquently described in the Introduction, the current study for the first time describes an alternative ALK5 mediated signaling pathway downstream of GDF3 in adipocytes that explains the improvement in metabolic health observed by knockdown of GDF3 in diet-induced obese adult mice. This obviously has implications for improving type 2 diabetic outcomes through targeting this GDF3 that is elevated during obesity.

GDF3 was already known to bind ALK7 (ACVR1c) type 1 receptor, it was therefore expected that it would act like Activin E, another ALK7 ligand. KO of either Activin E or ALK7 increases lipolysis, promoting loss of body weight and fat mass but paradoxically causing insulin resistance. In contrast, the authors found that although GDF3 can bind ALK7, GDF KO has the opposite effect of activin E KO in obese mice. GDF3 KO decreased lipolysis but promoted insulin sensitivity with no effect on body weight. To explain this paradox, the authors delve deeper into the downstream signaling pathways of GDF3 and find that the predominant signaling type I receptor for GDF3 is the TGF β type I receptor, also termed ALK5, with far less GDF3 signaling routing via ALK7. They also show that GDF3 can bind three type 2 receptors, upstream of these type I receptors, namely BMPR2, ACVR2A, and ACVR2B. Whereas, GDF3 inhibits signaling via BMPR2 (data?), it stimulates signaling via ACVR2A, and ACVR2B through TGF β R1/ALK5.

Overall, this is a well written manuscript with novel and biomedically important findings.

Reviewer #2

(Remarks to the Author)

The authors have made efforts to address the key concerns raised in the initial review, and their detailed rebuttal letter

provides valuable context for their data. However, after carefully reviewing the revised manuscript and accompanying rebuttal, it is clear that several critical points remain unresolved, and the interpretation of the data appears overextended in certain areas. Below are specific comments and suggestions for further improvement:

1. The authors explained experimental challenges in determining the in vivo concentration of GDF3, citing issues with band size discrepancies in western blotting and antibody specificity in ELISA. To address these concerns, it is strongly recommended that the authors provide western blot data for GDF3 in the GDF3 KO model (GDF3 fl/fl vs. whole-body or adipocyte-specific knockout [AKO]). Such data would clarify GDF3 deficiency in the experimental models used and offer critical information regarding the detectability of GDF3.

2. The authors propose that GDF3 regulates lipolysis by modulating β 3-adrenergic receptor (Adrb3) expression. However, several aspects of this mechanism require further clarification and experimental validation:

2-1. To determine whether the observed effects are Adrb3-dependent, forskolin treatment should be used to directly activate adenylyl cyclase, bypassing the β 3-adrenergic receptor. This experiment should be conducted under both ex vivo and in vitro conditions to evaluate differences in lipolysis between WT and GDF3 KO models.

2-2. Does Adrb3 expression decrease in adipose tissue of GDF3 whole-body or AKO mice? This key question needs to be addressed to support the proposed mechanism.

2-3. While the authors suggest that GDF3 alone can increase Adrb3 expression (Figs. 3G, I, L, and 7E, F, G), most experiments were performed in the presence of isoproterenol (Iso). This creates ambiguity regarding whether GDF3 independently regulates Adrb3 expression. To resolve this, the authors should evaluate the effects of single and combined treatments of GDF3 and Iso on Adrb3 expression, assessing both mRNA and protein levels.

2-4. In Fig. 7E, the increase in Adrb3 protein levels with GDF3 treatment appears modest, raising concerns about whether this change is sufficient to drive the observed lipolytic effects. This discrepancy is especially noticeable when compared to the significant changes presented in Fig. 7G. It is critical to present results that accurately reflect protein abundance.

2-5. In Fig. 3I, Iso treatment increases Adrb3 expression in control cells but not in KO cells. Conversely, Iso treatment does not induce Adrb3 expression in control cells in Fig. 7F. These differences may arise from variations in cell characteristics or GDF3 expression between the models used (primary vs. immortalized cells). Clarification is needed.

2-6. Consider using GDF3 KO cells as the control in Fig. 7 and supplementing with recombinant GDF3. This approach would provide more definitive evidence for GDF3's role in regulating Adrb3 expression and lipolysis.

3. In Figs. 3A and B, basal lipolysis is significantly reduced in adipose tissue explants from GDF3 whole-body KO mice. This finding is intriguing, as defects in canonical lipolytic regulators such as HSL, ATGL, or ADRB3 typically affect stimulated lipolysis more prominently. Moreover, these differences in basal lipolysis are absent in adipose tissue explants from GDF3 AKO mice. The authors should provide an interpretation of the relationship between GDF3 and basal lipolysis within the context of their proposed model.

Reviewer #3

(Remarks to the Author)

The authors have thoroughly addressed all concerns previously raised and considered suggestions and corrections. Their revision has significantly improved the quality of the manuscript.

Version 1:

Reviewer comments:

Reviewer #2

(Remarks to the Author)

The authors have appropriately addressed most questions raised by the reviewers.

Response to Nature Communications Reviewer #2: **Acute regulation of adipose tissue lipolysis and insulin resistance by the TGF β superfamily protein GDF3**

Kotikalapudi et al.

Reviewers comments are in blue

Authors responses are in black

1. The authors explained experimental challenges in determining the in vivo concentration of GDF3, citing issues with band size discrepancies in western blotting and antibody specificity in ELISA. To address these concerns, it is strongly recommended that the authors provide western blot data for GDF3 in the GDF3 KO model (GDF3 fl/fl vs. whole-body or adipocyte-specific knockout [AKO]). Such data would clarify GDF3 deficiency in the experimental models used and offer critical information regarding the detectability of GDF3.

Reviewers Figure 1: testing antibodies against GDF3 in loss and gain-of-function experiments. We include here representative images testing the specificity of an anti-GDF3 antibody. The predicted molecular weight of the full-length Gdf3 protein is 41 kDa. **A)** Epididymal adipose tissue (eWAT) from mice on a high-fat diet and following tamoxifen treatment for global inducible Gdf3 knockout as described in the manuscript. Each lane represents tissue from a distinct animal. Blotting with an anti-GDF3 antibody reveals three bands, one band of which is absent in the Gdf3^{KO} mice. Beta-actin is used as a loading control. **B)** Cos7 cells express low levels of Gdf3. These cells were transfected with the indicated amount of a plasmid encoding mouse wild type Gdf3 cDNA. Western blotting for Gdf3 or beta-tubulin was performed. In these experiments a 41 kDa band disappears in loss of function; or appears in Gdf3 gain-of-function models.

Representative western blots

TGF β superfamily proteins are highly similar and share primary and secondary structural features. These features complicate the creation of specific antibodies. Failure to account for cross-reactivity between similar proteins led to a significant controversy in the field due to cross reactivity between an antibody for GDF11 interacting with GDF8 (Egerman...Glass Cell Metabolism 2016). We anticipate cross-reactivity may be a common problem and that antibodies for Gdf3 must be carefully examined.

At the reviewer's request, we are including two western blots from either mouse epididymal adipose tissue (eWAT) or from a cell line transfected with a Gdf3-expressing plasmid. The Gdf3 protein is predicted to contain 366 amino acids and have a molecular weight of 41,586 Da.

Rev. Fig 1A: Mouse adipose tissue (eWAT) from control Gdf3^{fl/fl} and Gdf3^{KO} mice: loss of function

Gdf3 specificity. In mouse eWAT we observe a band corresponding to the predicted molecular weight of Gdf3, 41 kDa. This band is largely absent in the adipose tissue of Gdf3^{KO} mice. By these criteria, we believe this band reflect bona fide immunoreactivity with GDF3.

Cross reactivity. However, it is also appreciated that in the control mice, this represents the 3rd most intense band on the blot. Immunoreactive bands at 65 and 30 kDa are also present. The 65 kDa band is absent in the Gdf3^{KO} samples. The identity of these immunoreactive bands is not known.

Rev. Fig 1B: Cos cells expressing a Gdf3 plasmid: gain of function.

In this cell line, the anti-GDF3 antibody detects five bands. When a plasmid encoding wild-type murine Gdf3 cDNA is introduced, a 6th band appears at approximately 41kDa.

In both cases, this GDF3 antibody does recognize GDF3 and multiple other bands. We are confident that our Gdf3^{KO} model does exhibit lower levels of GDF3 protein. However this antibody would not be appropriate for ELISA-based detection of GDF3 due to cross-reactivity with non-specific proteins.

Issues surrounding failed attempts to create a specific antibody

We have been working to generate a *specific* Gdf3 antibody for more than four years. This has included work with the Institute for Protein Innovation, a nonprofit at Harvard. In addition, we are working with two different commercial manufacturers of antibodies. Despite attempting multiple technology platforms, we have yet to find any antibody that is both high-affinity and wholly specific for Gdf3. We have tested every commercially available we could obtain. We continue to work on this issue.

Quantitative real-time PCR results

Due to the issues with the antibodies illustrated above, we employed quantitative PCR (q-PCR) to confirm the Gdf3 deficiency in our knockout models. We use gene-specific primers to measure the Gdf3 transcript by q-PCR. Melting curve analysis shows one clear peak. This is a strong indication that the Gdf3 mRNA is decreased in our Gdf3 knockout model (See Figure 3E below) .

We believe that these data demonstrate that our KO model does express less Gdf3 compared to controls.

Figure 3E (left). Primary adipocytes differentiated from the stromal vascular fractions (SVF) of iWAT of Gdf3^{fl/fl} and Gdf3^{fl/fl} ::RosaCre^{ERT2/-} mice and treated with 4-hydroxytamoxifen (4-OHT) to generate Gdf3^{fl/fl} and Gdf3^{KO} cells. Quantitation of *Gdf3* mRNA levels by q-PCR.

2. The authors propose that GDF3 regulates lipolysis by modulating β 3-adrenergic receptor (Adrb3) expression. However, several aspects of this mechanism require further clarification and experimental validation:

2-1. To determine whether the observed effects are Adrb3-dependent, forskolin treatment should be used to directly activate adenylyl cyclase, bypassing the β 3-adrenergic receptor. This experiment should be conducted under both ex vivo and in vitro conditions to evaluate differences in lipolysis between WT and GDF3 KO models.

This is an excellent suggestion and tests the main mechanistic models proposed. These data are significant enough to be already illustrated in our overview mechanism figure; **Figure 8** is reproduced here (red oval added for emphasis). The referring data are in **Figure 3K** also reproduced below.

If the reduced lipolysis in Gdf3 knockouts is due to limited expression of the $\beta 3$ -adrenergic receptor ($\beta 3$ -AR), then activating signaling downstream of the receptor should restore lipolysis. That is exactly the result that we observe. The reviewer suggested using the adenylyl cyclase activator forskolin to elevate cellular levels of cAMP. In our studies, we used the cell permeable cAMP analog 8-Br-cAMP to achieve the same result.

In these experiments, we observed decreased stimulated lipolysis in Gdf3 knockout (Gdf3^{KO}) adipocytes compared to control (Gdf3^{fl/fl}) cells (**Fig 3K: left**). These results are consistent with our findings *in vivo* and *ex vivo* (Figure 3A-3D).

Here we show that the defect in stimulated lipolysis is completely restored when we elevate cAMP levels within these cells (**Fig 3K: right**).

This result suggests that the decreased rates of lipolysis seen in the Gdf3^{KO} model is due to a decrease in signaling upstream of cAMP production—consistent with a decrease in $\beta 3$ -AR receptor expression. This is further illustrated in Figure 3J where cellular levels of cAMP are lower in Gdf3^{KO} cells. We have graphically summarized these findings in **Figure 8**.

These results suggest that by altering the expression of the adipose tissue $\beta 3$ -adrenergic receptor (Adrb3), activin-type ligands like Gdf3 may play a critical role in maintaining glucose homeostasis via adipose tissue.

Figure 8. Graphical representation of Gdf3's action on adipocytes. Gdf3 regulates expression of $\beta 3$ -AR. Reduced $\beta 3$ -AR expression is overcome by elevating cellular levels of cAMP (red circle) to stimulate PKA activation of lipolytic enzymes.

Figure 3K. GDF3 loss of function leads to decreased lipolysis, an effect restored by increasing levels of cAMP. Primary adipocytes differentiated from the stromal vascular fractions (SVF) of iWAT of Gdf3^{fl/fl} and Gdf3^{fl/fl}::RosaCre^{E^{ERT2}/-} mice and treated with 4-hydroxytamoxifen (4-OHT) to generate Gdf3^{fl/fl} and Gdf3^{KO} cells. Relative fold change of FFA release in response to isoproterenol or 8-Bromo-cAMP (a cell-permeable cAMP analog) at the indicated time points.

2-2. Does *Adrb3* expression decrease in adipose tissue of GDF3 whole-body or AKO mice? This key question needs to be addressed to support the proposed mechanism.

We assessed the expression of *Adrb3* in our *Gdf3* knockout mice and noted a trend toward decreased expression in the epididymal white adipose tissue (eWAT). However, this decrease was not statistically significant compared to the control group in the eWAT as shown in **Reviewers' Figure 2**. It should be noted that these samples were collected from mice fasted for 4hr. Our in vitro studies show differences in *Adrb3* levels with pathway activation (see 3I below). It is possible that a 4hr fast is insufficient to stimulate *Adrb3* mRNA levels.

Reviewers' Figure 2: *Adrb3* mRNA levels in eWAT.

Adrb3 mRNA levels in eWAT epididymal adipose tissue (eWAT) from mice on a high-fat diet and with global inducible *Gdf3* knockout as described in the manuscript (see Figure 1). mRNA levels were not significantly different between genotypes.

2-3. While the authors suggest that GDF3 alone can increase *Adrb3* expression (Figs. 3G, I, L, and 7E, F, G), most experiments were performed in the presence of isoproterenol (Iso). This creates ambiguity regarding whether GDF3 independently regulates *Adrb3* expression. To resolve this, the authors should evaluate the effects of single and combined treatments of GDF3 and Iso on *Adrb3* expression, assessing both mRNA and protein levels.

We respectfully draw the reviewer's attention to an important aspect of our study: the experiments were conducted at different time points after isoproterenol treatment (**Figures 3G, I, 7E, and F**). In all the figures, "time 0" indicates the baseline lipolysis observed in the absence of isoproterenol. This observation suggests that *Gdf3* may be involved in priming the cells for isoproterenol stimulation, potentially enhancing lipolysis or regulating it through activin-type ligands. We do not claim that GDF3 alone can increase *Adrb3* expression, but rather suggest that GDF3 is one component of the regulatory process.

Figure 3G, 3I. Protein and mRNA levels of β 3-AR: Loss of function. Primary adipocytes differentiated from the stromal vascular fractions (SVF) of iWAT of *Gdf3^{fl/fl}* and *Gdf3^{fl/fl}::RosaCre^{ERT2/-}* mice and treated with 4-hydroxytamoxifen (4-OHT) to generate *Gdf3^{fl/fl}* and *Gdf3^{KO}* cells. **G**) Western blot analysis of β 3-AR before or following lipolytic stimulation with isoproterenol for indicated time points. β -actin levels serve as the loading control. **I**) Relative mRNA levels of *Adrb3* in response to isoproterenol stimulation for the indicated time points.

Figure 7E, 7F: Protein and mRNA levels of β 3-AR: Gain of function. Immortalized murine iWAT SVF cells were cultured and differentiated into mature adipocytes. **E**) Western blot analysis of cultured adipocytes treated with serum free media with or without 500ng/mL of rmGDF3 for 24 hours followed by isoproterenol stimulation for the indicated time points. Pictured are levels of β 3-AR and β -actin levels. **F**) Cultured adipocytes were maintained in serum free media (ctrl) or in serum free media with rmGDF3 (500 ng/mL) for 24 hours. Relative mRNA levels of *Adrb3* before or following isoproterenol stimulation for the indicated time points.

2-4. In Fig. 7E, the increase in *Adrb3* protein levels with GDF3 treatment appears modest, raising concerns about whether this change is sufficient to drive the observed lipolytic effects. This discrepancy is especially noticeable compared to the significant changes presented in Fig. 7G. It is critical to present results that accurately reflect protein abundance.

The modest differences in *Adrb3* expression observed in the western blot prompted us to use an alternative verification technique. Through immunocytochemistry, we clearly demonstrate the changes in *Adrb3* expression among the *Gdf3* gain- and loss of function models.

In the *Gdf3*^{KO} loss-of-function model we see similar levels of β 3-AR immunofluorescence at time zero in both genotypes. However, after 5 minutes of lipolytic stimulation β 3-AR levels are increased in the control cells but not in the *Gdf3*^{KO} (**Rev. Fig 3A**). In the gain-of-function model, treatment with recombinant GDF3 increases β 3-AR levels after 5 minutes of lipolytic stimulation (**Rev. Fig 3B**).

In this manuscript, we do not delve into how GDF3 regulates *Adrb3* mRNA levels, nor do we claim to explain how GDF3 regulates *Adrb3*. However there is ample evidence to suggest this is a common feature of the pathway. TGF β Type 1 receptors that activate SMAD2/3 signaling (ALK4/ALK5/ALK70029 also regulate *Adrb3* mRNA. These reports are consistent with our findings.

The dramatic *Adrb3* mRNA increase and modest protein expression change we see is accompanied by a clear phenotypic change (increased lipolysis with rmGDF3) that is reversed in the loss of function model.

2-5. In Fig. 3I, Iso treatment increases *Adrb3* expression in control cells but not in KO cells. Conversely, Iso treatment does not induce *Adrb3* expression in control cells in Fig. 7F. These differences may arise from variations in cell characteristics or GDF3 expression between the models used (primary vs. immortalized cells). Clarification is needed.

We appreciate your consideration of this detail in our findings. The two models presented show distinct differences: one exhibits a loss of function, while the other demonstrates a gain of function. Since these experiments were conducted separately—using either primary adipocytes or immortalized mature adipocytes—the normalization of expression was performed independently for each model. This approach contributes to the unique expression patterns observed in the results, which arise from the differences in the cell types and recombinant protein (rmGdf3) used in the study.

Reviewer's Figure 3: Effects of GDF3 on β 3-AR protein levels in adipocytes. **A**) Loss of function model. Primary adipocytes were differentiated from the stromal vascular fractions (SVF) of iWAT from *Gdf3*^{fl/fl} and *Gdf3*^{fl/fl}::*RosaCre*^{ERT2/-} mice. Once fully differentiated, cells were treated with 4-hydroxytamoxifen (4-OHT) to generate *Gdf3*^{fl/fl} and *Gdf3*^{KO} cells. Confocal images show immunofluorescence staining of β 3-AR (red) at baseline (0 minutes) and after lipolytic stimulation with isoproterenol. DAPI (blue) was used to stain the nuclei. **B**) Gain of function model. Immortalized murine iWAT SVF cells were cultured and differentiated into mature adipocytes. Confocal IF images staining β 3-AR (red) were captured at baseline (0 minutes) and following lipolytic stimulation (5 minutes) with isoproterenol, with DAPI (blue) utilized for nuclear staining. Cells were incubated with *Gdf3* 500ng/ml recombinant protein for 24 hours in serum-free media before lipolytic stimulation.

2-6. Consider using GDF3 KO cells as the control in Fig. 7 and supplementing with recombinant GDF3. This approach would provide more definitive evidence for GDF3's role in regulating ADRB3 expression and lipolysis.

We appreciate the reviewer's suggestion; however, we encountered significant technical challenges when we attempted the proposed experiment.

We differentiated primary adipocytes into mature adipocytes then induce the Gdf3 knockout with tamoxifen (4OHT) treatment. To measure the effect of exogenous ligands including Gdf3, our studies require serum starvation. This is an essential step because fetal bovine serum contains high levels of growth factors and TGF β superfamily ligands that would otherwise confound experimental interpretation. While the immortalized adipocyte cell lines tolerate the serum starvation, the primary Gdf3^{fl/fl} and Gdf3^{KO} adipocytes responded to the serum starvation by floating off of the dish. We were unable to effectively measure lipolysis due to these constraints. So, while we agree that it would be an informative study to combine both the loss and gain of function studies into the same experiment, there are significant technical limitations that prevent us from answering this question.

3. In Figs. 3A and B, basal lipolysis is significantly reduced in adipose tissue explants from GDF3 whole-body KO mice. This finding is intriguing, as defects in canonical lipolytic regulators such as HSL, ATGL, or ADRB3 typically affect stimulated lipolysis more prominently. Moreover, these differences in basal lipolysis are absent in adipose tissue explants from GDF3 AKO mice. The authors should provide an interpretation of the relationship between GDF3 and basal lipolysis within the context of their proposed model.

We would like to highlight some key findings from Fig. 3A, which examines basal lipolysis in inguinal white adipose tissue (iWAT). Our observations showed a significant decrease in both basal and stimulated free fatty acid (FFA) release in GDF3 knockout (GDF3^{KO}) mice. Notably, this decrease in basal lipolysis was not observed in epididymal white adipose tissue (eWAT). It is important to note that all tissue explants were collected following a 4-hour fasting period. This fasting can lead to dysregulation of the sympatho-adrenergic pathway in adipose tissue under a high-fat diet or obesity, which may affect basal lipolysis.

In contrast, our observations indicate no significant difference in the basal epididymal white adipose tissue (eWAT) of GDF3^{KO} mice, as shown in Fig. 3B. This non-significant trend is also consistent in Gdf3 adipocyte-specific knockout (Gdf3^{AKO}) mice, both for iWAT (Fig. 3C) and eWAT (Fig. 3D). Additionally, it's important to note that this lack of a significant effect on FFA release persists in our Gdf3 knockout mature adipocytes, as depicted in Fig. 3F. These findings suggest the potential role of Gdf3 in stimulating lipolysis during fasting or under other activating conditions. Additionally, we observed that Gdf3^{-AKO} mice, when subjected to cold stimulation, tend to utilize glucose as their primary energy source rather than fatty acids, which leads to better glucose homeostasis. This observation supports the idea that Gdf3 may facilitate lipolysis in the context of obesity, where both basal and stimulated lipolysis are often compromised.